# A REWARD-FREE VIEWPOINT ON MULTI-OBJECTIVE REINFORCEMENT LEARNING

**Ying-Tu Chen**[1,†] **Wei Hung**[1,†] **Bing-Shu Wu**[1,†] **Zhang-Wei Hong**[2] **Ping-Chun Hsieh**[1]

[1]National Yang Ming Chiao Tung University, Hsinchu, Taiwan
[2]Massachusetts Institute of Technology
{shu0924.cs13,pinghsieh}@nycu.edu.tw, zwhong@mit.edu

## ABSTRACT

Many sequential decision-making tasks involve optimizing multiple conflicting objectives, requiring policies that adapt to different user preferences. In multi-objective reinforcement learning (MORL), one widely studied approach addresses this by training a single policy network conditioned on preference-weighted rewards. In this paper, we explore a novel algorithmic perspective: leveraging reward-free reinforcement learning (RFRL) for MORL. While RFRL has historically been studied independently of MORL, it learns optimal policies for any possible reward function, making it a natural fit for MORL's challenge of handling unknown user preferences. We propose using the RFRL's training objective as an auxiliary task to enhance MORL, enabling more effective knowledge sharing beyond the multi-objective reward function given at training time. To this end, we adapt a state-of-the-art RFRL algorithm to the MORL setting and introduce a preference-guided exploration strategy that focuses learning on relevant parts of the environment. Through extensive experiments and ablation studies, we demonstrate that our approach significantly outperforms the state-of-the-art MORL methods across diverse MO-Gymnasium tasks, achieving superior performance and data efficiency. This work provides the first systematic adaptation of RFRL to MORL, demonstrating its potential as a scalable and empirically effective solution to multi-objective policy learning.[*]

## 1 INTRODUCTION

Many sequential decision-making tasks require optimizing multiple, often conflicting objectives. For example, in robot control, there is a trade-off between minimizing energy consumption and maximizing speed. One common approach to find a Pareto optimal policy is to maximize a weighted sum of the objectives, where the weights represent *user preferences*. User preferences depend on context—for instance, prioritizing speed in emergencies and energy efficiency in routine operations. Since user preferences are unknown in advance, solving multi-objective decision-making requires learning a set of policies for different preferences before testing.

Reinforcement learning (RL) (Sutton & Barto, 2018) has achieved strong performance in sequential decision-making, making multi-objective RL (MORL) a widely studied approach for learning policies for different user preferences (Hayes et al., 2022). A naive but inefficient solution is to train a separate policy for each preference. Another more scalable approach is to train a single policy network (Yang et al., 2019; Basaklar et al., 2023; Hung et al., 2023) conditioned on preferences, enabling parameter sharing and generalization across preferences. During training, the policy is optimized over a range of sampled preferences, each defining a reward function weighted by the preference. At test time, users can specify a preference to obtain the corresponding policy.

Another approach to handling unknown user preferences at test time is reward-free reinforcement learning (RFRL) (Jin et al., 2020; Touati et al., 2023), which has historically been developed independently of MORL despite addressing a similar problem. In RFRL, the agent explores the environment without receiving reward signals during training and instead learns a set of optimal policies

---

[†]Equal contribution.
[*]https://rl-bandits-lab.github.io/MORL-FB/

for any possible reward function in the environment. MORL can be seen as a special case of RFRL (Alegre et al., 2022), as RFRL does not restrict the reward function to be a weighted sum of pre-defined reward functions. However, despite their similarities, no prior work has explicitly adapted RFRL methods to solve MORL problems.

In this paper, we ask: *Can RFRL inform MORL?* We hypothesize that the objective of RFRL to learn optimal policies for any reward function could serve as a useful *auxiliary task* for MORL (Jaderberg et al., 2016; Rafiee et al., 2022; Veeriah et al., 2019). Although MORL under linear scalarization only needs optimal policies for linear combinations of known objectives, learning beyond these combinations could accelerate MORL via effective knowledge sharing. To investigate this question, we adapt a state-of-the-art RFRL algorithm (Touati et al., 2023) to the MORL setting, treating the preference-weighted reward as the test-time reward function given to RFRL. However, this naive approach performs poorly compared to existing MORL methods, likely because purely reward-free exploration does not prioritize states that are important for optimizing the preference-weighted reward in the given MORL task. As a result, the policies learned by RFRL for these reward functions could be suboptimal. To address this, we propose *guiding exploration using sampled preferences and mini-batch sampling*, directing the agent to visit states that maximize the corresponding preference-weighted reward function. This ensures that learning is focused on policies most relevant to MORL.

We highlight the main technical novelty of this paper: (1) **A new perspective of solving MORL**: We identify the close connection between MORL and RFRL, which have evolved independently despite tackling similar challenges of unknown user preferences at test time. This insight motivates new MORL algorithms by rethinking policy learning with multiple objectives from the perspective of RFRL. (2) **Algorithmic enhancements for adapting RFRL to MORL**: Even though RFRL and MORL are closely related, vanilla RFRL can perform poorly in the MORL setting (see Section 4). To address this, we introduce three key enhancements: (i) *Preference-guided exploration*: We propose to use the preference vector to sample latent vectors aligned with the target rewards to facilitate exploration in the latent space; (ii) *Training on latent vectors computed by mini-batch sampling from replay buffer as auxiliary tasks*: Our approach trains the policy network on latent vectors computed from mini-batch transitions sampled from the replay buffer. This design learns a broader range of policies than required for MORL and can be beneficial by providing auxiliary tasks; (iii) *Auxiliary Q loss*: To better adapt RFRL to MORL, we further facilitate the learning of representations from the observed reward vectors (instead of pseudo rewards as in RFRL) via an auxiliary Q loss as an additional learning signal.

Our experimental results demonstrate that our approach is both simple and effective. First, our method significantly outperforms the state-of-the-art MORL algorithms across various tasks in the MO-Gymnasium benchmark suite (Felten et al., 2023), including discrete and continuous control. Second, when trained with a limited number of preference samples, our method achieves substantially higher performance than other MORL approaches. This highlights that decoupling environment knowledge from reward information enhances generalization, particularly in scenarios with limited preference samples. To the best of our knowledge, this is the first work to adapt RFRL for MORL and present a practical algorithm that performs well across diverse deep RL tasks.

## 2 PRELIMINARIES

This section provides a brief review of MORL, along with the notation used throughout the paper. We use boldface symbols for vectors and matrices. For any $n \in \mathbb{N}$, we use $[n]$ as a shorthand for $\{1, \cdots, n\}$. For a set $\mathcal{Z}$, we let $\Delta(\mathcal{Z})$ denote the set of all probability distributions over $\mathcal{Z}$.

We formulate the MORL problem as an Multi-Objective Markov Decision Process (MOMDP) defined by the tuple $(\mathcal{S}, \mathcal{A}, \mathcal{P}, \mathbf{R}, \gamma, \mu)$, where $\mathcal{S}$ and $\mathcal{A}$ are the state and action spaces, $\mathcal{P} : \mathcal{S} \times \mathcal{A} \to \Delta(\mathcal{S})$ is the transition function, $\mathbf{R} : \mathcal{S} \times \mathcal{A} \to \mathbb{R}^d$ is a vector-valued reward function of $d$ objectives, $\gamma \in [0, 1)$ is the discount factor, and $\mu$ is the initial state distribution. Let $\Pi$ denote the set of all stationary randomized policies. Let $s_t, a_t, \boldsymbol{r}_t$ be the state, action, and reward received at time $t$. For a policy $\pi \in \Pi$, define $\mathbf{V}^\pi := \mathbb{E}_{\pi, s_0 \sim \mu}[\sum_{t=0}^\infty \gamma^t \mathbf{R}(s_t, a_t)]$ as the expected total discounted return vector achieved by $\pi$. Let $\mathbf{V}_i^\pi$ denote the $i$-th entry of $\mathbf{V}^\pi$. For a pair of policies $\pi$ and $\pi'$, we say that $\pi$ *Pareto-dominates* $\pi'$ (denoted by $\pi \succ \pi'$) if $\mathbf{V}_i^\pi \geq \mathbf{V}_i^{\pi'}$ for all $i \in [d]$ and there exists some $j \in [d]$ such that $\mathbf{V}_j^\pi > \mathbf{V}_j^{\pi'}$.

The general goal of MORL is to discover the *Pareto front*, which is defined as the set of non-dominated policies. That is, for each policy $\pi$ in the Pareto Front, there exists no other policy $\pi' \in \Pi$ such that $\pi' \succ \pi$. To search for the Pareto front, one common approach is to leverage a scalarization utility function $f_{\boldsymbol{\lambda}} : \mathbb{R}^d \to \mathbb{R}$ under a user preference vector $\boldsymbol{\lambda} \in \Lambda$, where $\Lambda$ denotes the preference set. In this paper, we focus on the linear scalarization setting where $f_{\boldsymbol{\lambda}}(\boldsymbol{r}) = \boldsymbol{\lambda}^\top \boldsymbol{r}$, as commonly adopted in the MORL literature (Abels et al., 2019; Yang et al., 2019; Basaklar et al., 2023; Hung et al., 2023; Lu et al., 2023). Without loss of generality, we presume that $\Lambda$ is the $d$-dimensional unit simplex. Notably, it has recently been shown by Lu et al. (2023) that any point on the Pareto front can be achieved by training a policy using linear scalarization due to the convexity of the policy-induced value function's range. Since the preference $\boldsymbol{\lambda}_{\text{test}}$ at test time is unknown during training, our goal is to learn a preference-conditioned policy $\pi : \mathcal{S} \times \Lambda \to \Delta(\mathcal{A})$ that can maximize the total discounted scalarized reward $\mathbb{E}\left[\sum_{t=1}^\infty \gamma^t \boldsymbol{\lambda}^\top \boldsymbol{r}_t\right]$, for *any* $\boldsymbol{\lambda} \in \Lambda$.

# 3 REWARD-FREE RL FOR MULTI-OBJECTIVE RL

This section explains why MORL can be seen as a special case of RFRL. We then discuss how this perspective enhances MORL by improving generalization and sample efficiency.

**MORL as a special case of RFRL.** The goal of RFRL is to compute an optimal policy for *any* scalar reward function $R : \mathcal{S} \times \mathcal{A} \to \mathbb{R}$ provided at test time, *without* observing any reward signal during training (*i.e.,* "reward-free") nor requiring additional environment interaction at test time. Formally, RFRL solves the following optimization problem at test time: $\arg\max_\pi \mathbb{E}_{\pi, s_0 \sim \mu}\left[\sum_{t=0}^\infty \gamma^t r_t\right]$, where $r_t$ is the reward realization that corresponds to the test-time reward function.

MORL under linear scalarization addresses a similar problem, but presumes the vector-valued reward signal from $\mathbf{R}(s, a)$ can be observed during training, and assigns $R(s, a) = \boldsymbol{\lambda}^\top \mathbf{R}(s, a)$, where $\boldsymbol{\lambda}$ (a user-specified preference vector) defines a linear combination of multiple reward components in $\mathbf{R}$. Both RFRL and MORL aim to retrieve an optimal policy for a given reward function at test time, but their approaches differ. While MORL typically focuses on finding the Pareto front by learning a set of optimal policies for various preferences $\boldsymbol{\lambda}$, RFRL learns policies for *all* possible reward functions, potentially including optimal policies for scalarized MORL rewards. RFRL achieves this by training a conditional policy network (Touati & Ollivier, 2021) or leveraging a pre-collected dataset through planning or batch RL (Jin et al., 2020), providing a broader policy set than traditional MORL.

**Key idea: RFRL as a source of auxiliary tasks.** RFRL learns policies for a broader class of reward functions than required for MORL, but this can be beneficial by providing *auxiliary tasks*. Prior research has shown that incorporating auxiliary tasks improves sample efficiency and generalization in RL (Jaderberg et al., 2016; Veeriah et al., 2019; Rafiee et al., 2022). Since RFRL naturally trains policies across a spectrum of reward functions, it provides a structured way to design these auxiliary tasks. However, directly applying RFRL to MORL can be data-inefficient since reward-free exploration may not prioritize states that are crucial for learning the Pareto front in MORL. The key challenge is: *How to utilize the auxiliary tasks offered by RFRL effectively to accelerate the learning of optimal policies in MORL?* In the sequel, we describe how RFRL can be adapted to effectively improve training in MORL.

## 3.1 FORWARD-BACKWARD MORL (MORL-FB)

In this section, we formally present MORL-FB by describing the implementation of RFRL and the key components that adapt RFRL to MORL by enhancing its learning efficiency.

**RFRL Implementation.** We implement RFRL for MORL using the state-of-the-art Forward-Backward (FB) RL algorithm (Touati & Ollivier, 2021). The FB method learns a set of policies optimized for different reward functions by decomposing the Q-value of an optimal policy for a scalar reward function $R$ into two neural networks: $\mathbf{F}_\theta$ (forward representation) and $\mathbf{B}_\omega$ (backward representation), where $\theta$ and $\omega$ denote their parameters. This decomposition allows the Q-function for a given reward function $R$ to be expressed as:

$$Q(s, a, \boldsymbol{z}_R) = \mathbf{F}_\theta(s, a, \boldsymbol{z}_R)^\top \boldsymbol{z}_R, \tag{1}$$

where $\boldsymbol{z}_R \in \mathbb{R}^{d_z}$ is an $d_z$-dimensional latent vector, and both $\mathbf{F}_\theta$ and $\mathbf{B}_\omega$ are neural networks producing $d_z$-dimensional outputs. Intuitively, $\boldsymbol{z}_R$ is meant to encode the optimal policy that corresponds to the current reward function. Once a reward function $R$ is revealed, $\boldsymbol{z}_R$ is defined as:

$$\boldsymbol{z}_R = \mathbb{E}_{(s,a)\sim\mathcal{D}}\left[\mathbf{B}_\omega(s,a)R(s,a)\right], \tag{2}$$

where $\mathcal{D}$ represents an arbitrary state-action distribution. In our implementation, we use $\mathcal{D}$ as the distribution induced by the replay buffer collected by the agent during training. Using this formulation, the greedy policy for a given reward function $R$ is defined as:

$$\pi(s, \boldsymbol{z}_R) = \arg\max_a \ \mathbf{F}_\theta(s, a, \boldsymbol{z}_R)^\top \boldsymbol{z}_R. \tag{3}$$

Now we are ready to present the design of MORL-FB at both training time and test time.

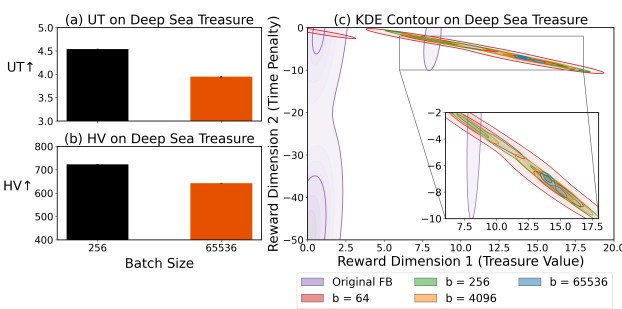

Figure 1: A motivating experiment on Deep Sea Treasure. (a)(b) Training performance (UT and HV defined in the sequel) of MORL-FB under different batch sizes for $\hat{\boldsymbol{z}}_{\boldsymbol{\lambda}}$. (c) KDE contour of return vector distributions of $\pi(\cdot, \boldsymbol{z})$ induced by $\hat{\boldsymbol{z}}_{\boldsymbol{\lambda}}$ (with various batch sizes $b$) and $\hat{\boldsymbol{z}} \sim \mathcal{N}(0, \mathbb{I}^{d_z})$. This shows that $\hat{\boldsymbol{z}}_{\boldsymbol{\lambda}}$ corresponds to learning for more diverse and relevant behavior in MORL than $\boldsymbol{z}_{\boldsymbol{\lambda}}$ and the $\boldsymbol{z}$ sampling strategy of the original FB. The detailed configuration is provided in Appendix C.

**Algorithm 1** MORL-FB

1: **Input:** $\boldsymbol{z}$ dimension $d_z$, sample number $n_s$
2: Initialize replay buffer $\mathcal{M} \leftarrow \varnothing$
3: **for** each iteration $i$ **do**
4:     Sample preference $\boldsymbol{\lambda}$ uniformly from $\Lambda$
5:     $\boldsymbol{z} \leftarrow$ PG-EXPLORE$(\boldsymbol{\lambda})$
6:     Generate rollouts using $\boldsymbol{z}$
7:     Sample $n_s$ transitions $\mathcal{D} \sim \mathcal{M}$
8:     Update FB networks $\mathbf{F}_\theta$, $\mathbf{B}_\omega$ and policy $\pi$
9: **end for**
10: **function** PG-EXPLORE$(\boldsymbol{\lambda})$
11:     Sample $n_s$ transitions $\mathcal{D} \sim \mathcal{M}$
12:     $\boldsymbol{z} \leftarrow \sum_{(s,a,\mathbf{r},s')\in\mathcal{D}} \frac{\mathbf{B}_\omega(s,a)\mathbf{r}^\top\boldsymbol{\lambda}}{n_s}$
13:     Normalize $\boldsymbol{z}$ to $\boldsymbol{z} \leftarrow \sqrt{d_z}\frac{\boldsymbol{z}}{\|\boldsymbol{z}\|_2}$
14:     **return** $\boldsymbol{z}$
15: **end function**

**Test Time:** At test time, we can easily adapt Equation (2) by replacing $R(s, a)$ with a user-specified scalarized multi-objective reward based on a preference vector $\boldsymbol{\lambda}$ as $R(s, a) = \boldsymbol{\lambda}^\top \mathbf{R}(s, a)$. Next, given the learned $\mathbf{F}_\theta$ and $\mathbf{B}_\omega$, we compute the corresponding $\boldsymbol{z}_R$ and use it in the policy defined by Equation (3). This effectively retrieves an optimal policy for the given preference $\boldsymbol{\lambda}$.

**Training with Preference-Guided Exploration:** During training, $\mathbf{F}_\theta$, $\mathbf{B}_\omega$, and $\pi$ must be trained by sampling $\boldsymbol{z}$ and conditioning the networks on these sampled values. Since the test-time user preference $\boldsymbol{\lambda}_{\text{test}}$ is unknown at this stage, we cannot directly compute $\boldsymbol{z}$ using Equation (2). At a high level, training on a diverse set of $\boldsymbol{z}$ samples is equivalent to training the agent on a variety of reward functions, since $\boldsymbol{z}$ is inherently linked to rewards through Equation (2).

In principle, $\boldsymbol{z}$ can be sampled from any distribution without restriction. In (Touati & Ollivier, 2021), $\boldsymbol{z}$ is drawn from a standard normal distribution $\mathcal{N}(0, \mathbb{I}^{d_z})$ in a $d_z$-dimensional space. However, we found that this approach leads to poor sample efficiency when testing the agent on MORL tasks. We hypothesize that sampling $\boldsymbol{z}$ from a normal distribution produces representations that differ significantly from the actual $\boldsymbol{z}_R$ obtained from a preference-weighted multi-objective reward function (see Figure 5 for a visualization of empirical $\boldsymbol{z}$ distributions).

To address this issue, we propose *Preference-Guided Exploration* (PG-Explore), which constructs a more relevant $\boldsymbol{z}$ distribution via sampling guided by preference-weighted rewards. The design of PG-Explore builds on the following insights:

- **Using $\{\boldsymbol{z}_{\boldsymbol{\lambda}}\}_{\boldsymbol{\lambda}\in\Lambda}$ only leads to limited exploration of $\boldsymbol{z}$**: Recall that in MORL, we can observe multi-objective rewards $\mathbf{R}(s, a)$ (or its noisy version) during training. One direct approach is to compute $\boldsymbol{z}$ as:

$$z_{\boldsymbol{\lambda}} = \mathbb{E}[\mathbf{B}_\omega(s,a)\boldsymbol{\lambda}^\top \mathbf{R}(s,a)] \stackrel{(a)}{=} \mathbb{E}[\mathbf{B}_\omega(s,a)\mathbf{R}(s,a)^\top \boldsymbol{\lambda}] \stackrel{(b)}{=} \underbrace{(\mathbb{E}[\mathbf{B}_\omega(s,a)\mathbf{R}(s,a)^\top])}_{=:\mathbf{H}} \boldsymbol{\lambda}, \quad (4)$$

where (a) holds due to the fact that $\boldsymbol{\lambda}^\top \mathbf{R}(s,a)$ is a scalar and can be swapped in the matrix multiplication with $\mathbf{B}_\omega(s,a)$ and (b) follows from that $\boldsymbol{\lambda}$ can be moved out of the expectation. Equation (4) suggests that $z_{\boldsymbol{\lambda}}$ *is in the span of $d$ preference-agnostic column vectors of the $d_z \times d$ matrix* $\mathbf{H}$, for any preference $\boldsymbol{\lambda}$. In practice, since the number of objectives $d$ is usually much smaller than $d_z$, the coverage of $\{z_{\boldsymbol{\lambda}}\}_{\boldsymbol{\lambda}\in\Lambda}$ in $\mathbb{R}^{d_z}$ can be extremely small. This leads to limited exploration of $z$ during training such that the agent can commit to a set of improper $z$, especially in the early training stage when $\mathbf{F}_\theta$ and $\mathbf{B}_\omega$ are not well trained.

- **Constructing $\hat{z}_{\boldsymbol{\lambda}}$ by mini-batch sampling for exploration**: To encourage exploration of $z$ relevant to MORL, we propose a conceptually simple and yet effective technique that leverages mini-batch sampling to construct $\hat{z}_{\boldsymbol{\lambda}}$. Specifically, we sample a batch of $n_s$ data samples (denoted by $\mathcal{D}$) from the replay buffer and compute $\hat{z}_{\boldsymbol{\lambda}} = \sum_{(s,a,\mathbf{r},s')\in\mathcal{D}} \mathbf{B}_\omega(s,a)\mathbf{r}^\top \boldsymbol{\lambda}/n_s$. Figure 1 shows a comparison of training with $z_{\boldsymbol{\lambda}}, \hat{z}_{\boldsymbol{\lambda}}$ under various batch sizes, and $z$ drawn from $\mathcal{N}(0,\mathbb{I}^{d_z})$ as in the original FB method, in Deep Sea Treasure (DST), which is a goal-oriented navigation task with two-dimensional rewards as (treasure value, step cost). As the true $z_{\boldsymbol{\lambda}}$ is not available, we use $\hat{z}_{\boldsymbol{\lambda}}$ with a large $n_s$ as a surrogate for $z_{\boldsymbol{\lambda}}$. We can see that: (i) $\hat{z}_{\boldsymbol{\lambda}}$ indeed corresponds to learning more diverse behavior than just learning for $z_{\boldsymbol{\lambda}}$. (ii) $\hat{z}_{\boldsymbol{\lambda}}$'s are more relevant to the reward functions in MORL encountered at test time than the $z$ sampled from $\mathcal{N}(0,\mathbb{I}^{d_z})$, improving sample efficiency.

- **Learning induced by $\hat{z}_{\boldsymbol{\lambda}}$ serves as auxiliary tasks:** Recall that in PG-Explore, we construct $\hat{z}_{\boldsymbol{\lambda}} = \sum_{(s,a,\mathbf{r},s')\in\mathcal{D}} \mathbf{B}_\omega(s,a)\mathbf{r}^\top \boldsymbol{\lambda}/n_s$ by mini-batch sampling. This means that for any given preference $\boldsymbol{\lambda}$, the agent can learn beyond $z_{\boldsymbol{\lambda}}$ and, moreover, from multiple values $z$ from different batches of transitions, providing richer learning signals. This approach is closely related to the auxiliary tasks in deep RL, where training objectives that are not directly or totally aligned with the target objective have been shown to accelerate learning (Jaderberg et al., 2016; Veeriah et al., 2019; Rafiee et al., 2022).

**Training Objective Functions:** (i) *Measure loss*: To train the $\mathbf{F}_\theta$ and $\mathbf{B}_\omega$ networks in MORL-FB, we use the standard measure loss $\mathcal{L}_{\mathrm{M}}(\mathbf{F}_\theta, \mathbf{B}_\omega; z_{\boldsymbol{\lambda}})$, which minimizes the Bellman residual on the successor measure (Touati et al., 2023). $\mathbf{F}_{\bar{\theta}}$ and $\mathbf{B}_{\bar{\omega}}$ are target networks. This loss is defined as:

$$\mathcal{L}_{\mathrm{M}}(\mathbf{F}_\theta, \mathbf{B}_\omega; z_{\boldsymbol{\lambda}}) = \mathbb{E}_{\substack{(s_t,a_t,s_{t+1})\sim\mathcal{D} \\ (s',a')\sim\mathcal{D}}} \Big[ \big(\mathbf{F}_\theta(s_t,a_t,z_{\boldsymbol{\lambda}})^\top \mathbf{B}_\omega(s',a')$$
$$-\gamma\mathbf{F}_{\bar{\theta}}(s_{t+1},\pi(s_{t+1},z_{\boldsymbol{\lambda}}),z_{\boldsymbol{\lambda}})^\top \mathbf{B}_{\bar{\omega}}(s',a')\big)^2 \Big]$$
$$- 2\mathbb{E}_{(s_t,a_t,s_{t+1})\sim\mathcal{D}}[\mathbf{F}_\theta(s_t,a_t,z_{\boldsymbol{\lambda}})^\top \mathbf{B}_\omega(s_{t+1},a_{t+1})]. \quad (5)$$

(ii) *Auxiliary Q loss*: In the context of MORL, we propose to employ an auxiliary Q loss to facilitate the learning of FB representations from the *observed reward vectors*, instead of the *pseudo rewards* in the original FB (also see the ablation study in Section 4.1). Specifically, the Q-loss is constructed as the squared temporal difference error represented in $\mathbf{F}_\theta$ and $\mathbf{B}_\omega$, and the transitions are sampled from the replay buffer to compute $z_{\boldsymbol{\lambda}}$ via our preference-guided function:

$$\mathcal{L}_Q(\mathbf{F}_\theta; z_{\boldsymbol{\lambda}}) = \mathbb{E}_{(s,a,\mathbf{r},s')\sim\mathcal{D}}\big[(\mathbf{F}_\theta(s,a,z_{\boldsymbol{\lambda}})^\top z_{\boldsymbol{\lambda}} - (\boldsymbol{\lambda}^\top \mathbf{r} + \gamma\mathbf{F}_{\bar{\theta}}(s',\pi(s',z_{\boldsymbol{\lambda}}),z_{\boldsymbol{\lambda}})^\top z_{\boldsymbol{\lambda}}))^2\big]. \quad (6)$$

We summarize the implementation in Algorithm 1. The detailed pseudo code (Algorithm 2) and further details about loss functions and implementation are provided in Section B. Note that the FB framework can use either state-dependent or state-action-dependent backward representation, and both perform well in practice (see Appendix D.4). As the original FB (Touati et al., 2023) presumes a state-dependent design, we focus mainly on state-dependent ones in the subsequent experiments.

## 4 EXPERIMENTS

**Evaluation Domains.** We leverage the MO-Gymnasium benchmark suite (Felten et al., 2023) and consider various discrete and continuous control tasks. For the continuous control tasks, we consider

robot locomotion tasks in Multi-objective MuJoCo with up to 5 objectives, including Walker2d, Halfcheetah2d, Ant3d, Hopper3d, Humanoid2d, and Humanoid5d. Each environment presents a unique set of objectives, *e.g.,* the goal of Ant3d is to optimize both x-axis and y-axis speeds while minimizing energy consumption. The experiments of the discrete control tasks are provided in Appendix D.3. The detailed configurations for all tasks are provided in Appendix C.

**Benchmark Methods.** To evaluate the effectiveness of our proposed approach, we compare MORL-FB against various benchmark methods, including: (i) *Single preference-conditioned policy methods*: PD-MORL (Basaklar et al., 2023), Q-Pensieve (Hung et al., 2023), CAPQL (Lu et al., 2023), Envelope Q-Learning (EQL) (Yang et al., 2019), and PCN (Reymond et al., 2022); (ii) *Multi-policy MORL*: PG-MORL (Xu et al., 2020), SFOLS (Alegre et al., 2022), MORL/D (Felten et al., 2024), GPI-LS, and GPI-PD (Alegre et al., 2023) ; (iii) *Reward-free RL*: We take the original FB approach (Touati et al., 2023) as a baseline.

Regarding PD-MORL, Q-Pensieve, and FB, we leverage their official implementations from (Basaklar et al., 2023; Hung et al., 2023; Touati et al., 2023). To ensure a fair comparison among all the benchmark methods, we adopt the standard PD-MORL without the auxiliary pre-trained preference interpolator, which essentially requires a substantial amount of additional data for pre-training and could bias the comparison. Regarding CAPQL, EQL, PCN, PG-MORL, MORL/D, GPI-LS, and GPI-PD, we leverage the implementation of MORL-Baselines (Felten et al., 2023) for better reproducibility. As the PG-MORL in MORL-Baselines can only support two-objective tasks, we extend this code base to accommodate those tasks beyond two objectives. On the other hand, the original SFOLS only focuses on discrete control tasks by default. For a more thorough comparison, we utilize its official implementation for discrete problems and extend SFOLS with a TD3 backbone for evaluation on continuous control. Moreover, we apply hyperparameter optimization to the benchmark algorithms and MORL-FB. Appendix C.2 details the selection range of hyperparameters and the final selected values. For all the tasks, we run each algorithm for 3 million environment steps, which is comparable to most of the existing MORL studies. Below we report the average performance and the empirical standard deviation over 5 random seeds for each task. More detailed configurations of the experiments and benchmark methods are provided in Appendix C.

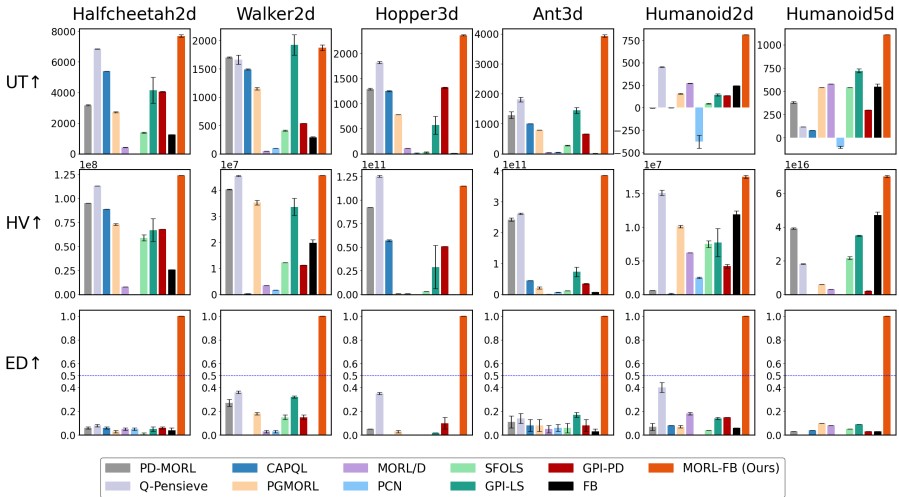

Figure 2: Evaluation of MORL-FB and several MORL benchmark algorithms on diverse continuous control tasks within the MO-Gymnasium suite, assessing performance using key metrics. These results demonstrate the clear advantage of MORL-FB across all tested benchmarks.

**Evaluation Metrics.** We evaluate the performance of each algorithm using three metrics that are widely used in the MORL literature (Van Moffaert & Nowé, 2014; Yang et al., 2019; Kyriakis & Deshmukh, 2022; Basaklar et al., 2023; Hung et al., 2023; Lu et al., 2023):

- **Utility (UT)**: To evaluate the scalarized total reward across different preferences at an aggregate level, we employ the utility metric defined as $\mathbb{E}_{\boldsymbol{\lambda}}[\sum_t \boldsymbol{\lambda}^\top \boldsymbol{r}_t]$, where the expectation is taken with respect to the uniform distribution over the preference set $\Lambda$ (*i.e., d*-dimensional unit simplex).

- **Hypervolume (HV)**: As a standard metric in the literature of general multi-objective optimization, hypervolume naturally captures the inherent trade-off among different objective functions using one aggregate scalar value (Zitzler & Thiele, 1999). Specifically, given a reference point $\boldsymbol{u}_{\text{ref}} \in \mathbb{R}^d$ and any collection for return vectors $\mathcal{U} \subseteq \mathbb{R}^d$, the hypervolume of $\mathcal{U}$ can be formally defined as $\text{HV}(\mathcal{U}; \boldsymbol{u}_{\text{ref}}) := \mu\Big( \bigcup_{\boldsymbol{u} \in \mathcal{U}} \{ \boldsymbol{y} | \boldsymbol{u} \succeq \boldsymbol{y} \succeq \boldsymbol{u}_{\text{ref}} \} \Big)$, where $\mu(\cdot)$ denotes the $d$-dimensional Lebesgue measure. In practice, $\boldsymbol{u}_{\text{ref}}$ is selected based on the range of possible total return and is task-dependent. The configuration of $\boldsymbol{u}_{\text{ref}}$ for each MORL task is provided in Appendix C.

- **Episodic Dominance (ED)**: As a metric complementary to HV and UT, ED is meant to capture the relative strength of a pair of algorithms under different preferences. Specifically, given any two algorithms $\texttt{ALG}_1, \texttt{ALG}_2$, we define $\text{ED}(\texttt{ALG}_1, \texttt{ALG}_2) := \mathbb{E}_{\boldsymbol{\lambda}}[\mathbb{I}\{\boldsymbol{\lambda}^\top \boldsymbol{g}(\tau_{\texttt{ALG}_1}) \geq \boldsymbol{\lambda}^\top \boldsymbol{g}(\tau_{\texttt{ALG}_2})\}]$, where $\boldsymbol{g}(\cdot)$ denotes the trajectory-wise cumulative return vector, $\tau_{\texttt{ALG}_1}$ and $\tau_{\texttt{ALG}_2}$ are the trajectories generated under the policies of $\texttt{ALG}_1$ and $\texttt{ALG}_2$, and $\boldsymbol{\lambda}$ is drawn uniformly from $\Lambda$. Note that we use 500 uniformly sampled preference vectors and evaluate across 5 distinct random seeds for each preference vector for statistical robustness.

To ensure rigorous evaluation, we further follow the guidelines of (Agarwal et al., 2021) by taking the normalized UT scores and reporting the aggregated performance across tasks in median, mean, and interquartile mean (IQM). Regarding the normalized scores, we follow the procedure in (Fu et al., 2020), which (i) employs a *random policy*—where actions are selected uniformly at random— as the baseline with normalized score of 0 and (ii) an *expert policy* trained by single-objective SAC—as the topline with normalized score of 100. The above normalization is done on a per-preference basis.

**Does the reward-free viewpoint of MORL-FB improve sample efficiency over the benchmark methods?** Figure 2 shows the performance of all the methods in UT, HV, and ED for continuous control tasks. Regarding ED, for each baseline algorithm ALG, we report $\text{ED}(\texttt{ALG}, \texttt{MORL-FB})$ to show the pairwise comparison. We can make the following observations: (i) MORL-FB achieves either the best or close to the best UT and HV among all methods on all the tasks, regardless of the number of objectives. This showcases that MORL-FB is indeed sample-efficient in the sense that it can discover a diverse collection of high-performing policies across various preferences using only as few as 3 million samples as used by the *expert policy*. (ii) Given that $\text{ED}(\texttt{ALG}, \texttt{MORL-FB})$ are consistently smaller than 0.5 for all baselines, we see that MORL-FB outperforms all benchmark methods (including the state-of-the-art methods like PD-MORL and Q-Pensieve), under most preferences. (iii) PD-MORL and Q-Pensieve perform well on two-objective tasks (*e.g.,* Halfcheetah2d and Walker2d) but underperform when the number of objectives is larger (*e.g.,* Ant3d and Humanoid5d). Additional results on discrete control environments can be found in Section D.3.

Moreover, regarding the aggregated results, Figure 3 shows that MORL-FB reliably outperforms the benchmark methods both in conventional statistics (*e.g.,* mean and median) and robust metrics like IQM. MORL-FB achieves the best IQM scores by a large margin *vis-à-vis* other methods, confirming the significant improvements over the state-of-the-art MORL.

Recall that MORL-FB leverages PG-Explore to address the fundamental exploration issue of vanilla FB, which suffers from sample inefficiency in MORL. Remarkably, the per-task results in Figure 2 and aggregated results in Figure 3 show that MORL-FB enjoys significantly better UT and HV across tasks. Accordingly, the ED scores $\text{ED}(\texttt{FB}, \texttt{MORL-FB})$ remain nearly zero in all tasks.

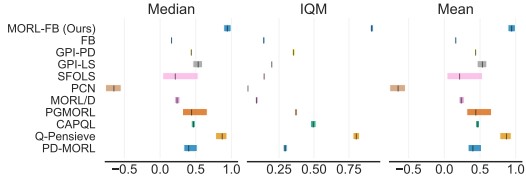

Figure 3: Evaluation of MORL-FB and several MORL benchmark algorithms using aggregate metrics, including median, mean, and interquartile mean (IQM). These results show the superior performance of MORL-FB across all metrics.

**Does MORL-FB achieve effective generalization across preferences?** To better assess the generalization capabilities of MORL-FB across preferences, we further evaluate the algorithms in a stylized setting where they are trained only on a small set of preference vectors $\Lambda_{\text{train}}$ (rather than the whole $\Lambda$) and aim for generalization over $\Lambda$ at test time. Specifically, for a $d$-objective task, we let $\Lambda_{\text{train}}$ include only the standard basis preferences, *i.e.,* $d$-dimensional one-hot vectors, and the uniform preference vector

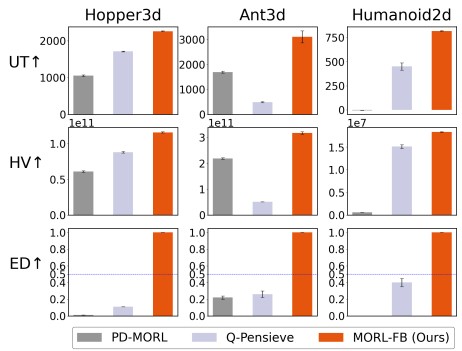

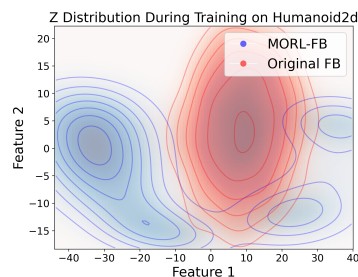

Figure 4: Evaluation of MORL-FB and benchmark methods (PD-MORL and Q-Pensieve) under a reduced preference set $\Lambda_{\text{train}}$ during training: These demonstrate the generalization capability of MORL-FB across preferences.

Figure 5: Empirical $z$ distribution under t-SNE for Humanoid2d with MORL-FB (preference-guided sampling, blue) and original FB (standard normal, red): The multi-modal distribution observed with MORL-FB suggests a more diverse set of latent representations compared to unimodal nature of original FB.

$[1/d, \cdots, 1/d]$. The testing setup is exactly the same as that for Figure 2. Here we focus on comparing MORL-FB to PD-MORL and Q-Pensieve, which are the top two benchmark methods in Figure 2 and utilize conditioned networks of structures different from MORL-FB.

As shown in Figure 4, both PD-MORL and Q-Pensieve exhibit a notable decline in performance across all three metrics compared to those in Figure 2. In contrast, MORL-FB maintains consistent performance across the evaluated tasks, with only minimal degradation in UT and HV values compared to Figure 2. These findings showcase that MORL-FB can generalize more effectively over the entire preference set, even when trained on a limited set of preference vectors. More detailed results, such as the numerical values and the aggregated performance (*e.g.,* IQM) are in Appendix D.

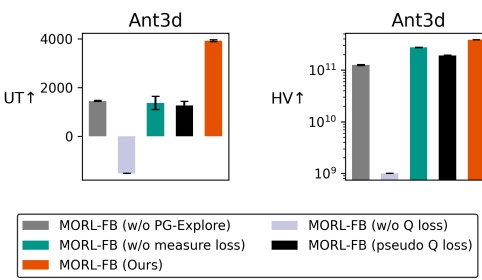

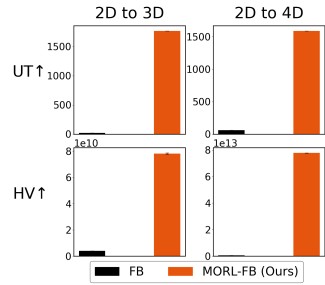

Figure 6: Evaluation of MORL-FB and its ablated versions on the Ant3d task. The results highlight the importance of PG-Explore and auxiliary losses, as removing these components leads to performance degradation.

Figure 7: Zero-shot cross-objective transfer from Hopper2d to Hopper3d and Hopper4d using vanilla FB and MORL-FB: Results demonstrate effective transfer by MORL-FB, supporting the efficacy of its proposed enhancements.

## 4.1 ABLATION STUDY

**Preference-Guided Exploration (PG-Explore).** To investigate the benefits of sampling $z$ from a preference-guided distribution, we perform an ablation study on comparing the proposed MORL-FB and a variant of MORL-FB that samples $z$ from $\mathcal{N}(0, \mathbb{I}^{d_z})$, *i.e.,* the distribution adopted by the vanilla FB. Figure 6 (specifically the bars in gray and orange) shows that MORL-FB indeed benefits significantly from a preference-guided distribution across the tested task, highlighting its importance in enabling directed exploration and sample-efficient policy learning.

Moreover, we visualize the empirical distributions of the sampled $z$ of MORL-FB and the original FB. Specifically, we record the $z$ vectors used throughout training and apply t-SNE (Van der Maaten & Hinton, 2008) for visualization in a two-dimensional space. The results on Humanoid2d in Figure 5 show that sampling $z$ from a normal distribution results in a unimodal empirical distribution (contours in red). By contrast, MORL-FB with the preference-guided sampling exhibits a multi-

modal distribution (contours in blue), indicating a richer and more diverse set of $z$ distributions. This multi-modality allows MORL-FB to better capture the underlying reward structure, achieving improved generalization and adaptation to various objectives. More visualization of $z$ distributions for other tasks can be found in Appendix D.

**Auxiliary Q loss.** To corroborate the efficacy of the auxiliary Q loss, we further conduct an ablation study on this term. From Figure 6 (specifically the bars in black and orange), the Q loss can facilitate the learning of forward and backward representations in MORL-FB and thereby boost the performance in both UT and HV. More ablation results across environments are in Appendix D.2.

## 4.2 ZERO-SHOT CROSS-OBJECTIVE TRANSFER

Recall that one salient feature of MORL-FB is to use $z$ to encode the $\lambda$-dependent scalarized reward function. Accordingly, MORL-FB is endowed with the ability to achieve zero-shot transfer even across tasks of different number of objectives. Such *zero-shot cross-objective transferability* allows us to add new factors to the reward function without the need for retraining and hence is a very useful feature in practice. To validate this, we use MORL-FB to learn the $\mathbf{F}$ and $\mathbf{B}$ networks on Hopper2d and directly evaluate them on Hopper3d and Hopper4d, which involve additional reward terms like "jump height" and "z-axis speed", at test time. Details of the environment configurations are in Appendix C. We conducted the same evaluation for vanilla FB (Touati et al., 2023) as a baseline. From Figure 7, vanilla FB cannot achieve effective cross-objective transfer given that vanilla FB already suffers in the standard MORL setting (cf. Figure 2). By contrast, MORL-FB achieves effective transfer across objectives in a zero-shot manner, corroborating the proposed enhancements.

## 5 RELATED WORK

### 5.1 MULTI-OBJECTIVE RL

**Single preference-conditioned policy methods.** Single preference-conditioned policy methods learn one policy network that adapts to different objective trade-offs by conditioning on preference. Many of these methods employ scalarization techniques (Van Moffaert et al., 2013; Yang et al., 2019), transforming multi-objective problems into weighted single-objective problems. They allow policies to dynamically adjust at inference time. However, relying solely on linear reward aggregation without proper representation learning can lead to suboptimal solutions. To address this, CAPQL (Lu et al., 2023) introduced the concave reward terms for better optimization landscapes, while CN-DER (Abels et al., 2019) proposed a preference-conditioned Q-network with an experience replay mechanism to handle dynamic weights and mitigate non-stationarity. Q-Pensieve (Hung et al., 2023) improved sample efficiency by reusing past policy snapshots. Without using scalarization, Abdolmaleki et al. (2020) proposed to learn action distributions per objective and fitted a parametric policy via supervised learning. To improve adaptability to diverse and unseen preference vectors, methods like PCN (Reymond et al., 2022) formulated MORL as a classification problem, and MOAC (Zhou et al., 2024) finds Pareto-stationary points by adapting multi-gradient descent to MORL without scalarization. PD-MORL (Basaklar et al., 2023) also trains a single preference-conditioned network but directly incorporates preference vectors, for example, through cosine similarity measures within its value-function update rule to efficiently learn a comprehensive set of policies across the continuous preference space.

**Multi-policy methods.** Multi-policy methods explicitly learn multiple policies to cover the Pareto front, capturing diverse trade-offs in the training process. A key challenge in multi-policy MORL is efficiently constructing a coverage set that represents the full Pareto front while maintaining scalability. To refine policy selection and handle dominated actions, Lizotte et al. (2012) introduced a structured approach using linear value function approximation. Subsequent methods (Kyriakis & Deshmukh, 2022; Van Moffaert & Nowé, 2014; Xu et al., 2020) focused on improving exploration efficiency across the preference space but lacked structured learning mechanisms to generalize across diverse preferences. Building on the idea of incorporating structure into policy learning, Felten et al. (2024); Mossalam et al. (2016) extended structured learning for multi-objective RL, employing decomposition and sequential single-objective optimization to enhance efficiency. However, scalability and adaptability remained challenges. DG-MORL (Lu et al., 2024) leveraged demonstrations and a self-evolving mechanism to improve scalability. As for improving adaptability, Mossalam et al. (2016) extended Optimistic Linear Support (OLS) to deep RL, constructing a

convex coverage set through a sequence of single-objective, providing a structured way to represent diverse trade-offs, but lacked effective transferability.

## 5.2 Successor Features for Transfer Across Reward Functions

One related transfer setting in RL is to learn policies for all reward functions that are linear combinations of a finite set of known features. Barreto et al. (2017) proposed the *successor feature* (SF), which reflects the state-action occupancy of a policy and can be viewed as an extension of the classic successor representation (Dayan, 1993). By design, SFs achieves transfer across reward functions in two ways: (i) For a fixed policy $\pi$, SFs can enable fast policy evaluation of $\pi$ across different reward functions; (ii) Given a set of policies, SFs can be combined with the generalized policy improvement (GPI) update to generate a new policy that is no worse than the given set of policies and has a sub-optimality gap characterized by the task differences, for any reward function (Barreto et al., 2017; 2018). Accordingly, SFs has been combined with deep neural networks and applied to various tasks like subgoal extraction (Kulkarni et al., 2016) and robot navigation (Zhang et al., 2017). Subsequently, Borsa et al. (2019) extended the SFs by decoupling the policy and task description for better representational flexibility. Moreover, Alegre et al. (2022) applied SFs and GPI to solve MORL by enabling transfer across different scalarized reward functions. Chua et al. (2024) proposed a practical method for learning the SFs directly from pixels for image-based control. More recently, Zhang et al. (2024) established the convergence analysis with generalization guarantees for SFs under neural function approximation.

Despite the transfer capability, the limitations of SFs and its variants are mainly two-fold: (i) SFs require a set of pre-defined reward features, which could miss important aspects of the environment and limit adaptability. (ii) While SFs can achieve fast policy evaluation across reward functions, SFs cannot directly induce an optimal policy when given an arbitrary new reward function at test time.

## 5.3 Reward-Free RL

Among the studies in the RFRL literature, (Touati & Ollivier, 2021; Touati et al., 2023) are the most relevant to our work. To address the aforementioned issues of SFs, Touati & Ollivier (2021) adopted the reward-free MDP formulation and proposed a low-rank model termed the forward-backward (FB) representations, which capture the state-action occupancy of the optimal policies for all the reward functions by learning the required features directly from data. The FB framework has been implemented and validated for both reward-free discrete control (Touati & Ollivier, 2021) and continuous control (Touati et al., 2023). Subsequently, the framework of FB representations has been extended to various settings, such as offline RL with low-quality data (Jeen et al., 2024), online unsupervised RL (Sun et al., 2025), imitation learning (Pirotta et al., 2024; Tirinzoni et al., 2025), and partially-observable MDPs (Jeen et al., 2025).

Another line of RFRL research focuses on achieving provably efficient exploration without using any reward information, in both tabular settings (Jin et al., 2020; Kaufmann et al., 2021; Ménard et al., 2021; Wu et al., 2022) and under function approximation (Qiu et al., 2021; Wagenmaker et al., 2022; Wang et al., 2020; Zanette et al., 2020; Zhang et al., 2021).

Inspired by the RFRL literature, we propose to rethink MORL via RFRL and adapt the FB method to boost the sample efficiency and generalization in MORL.

## 6 Conclusion, Limitations, and Future Work

We propose MORL-FB, which offers a new perspective by rethinking MORL through the lens of RFRL. By using RFRL as auxiliary tasks, MORL-FB provides the first systematic adaptation of RFRL to MORL and enhances its sample efficiency with critical algorithmic enhancements, including preference-guided exploration with mini-batch sampling and an auxiliary Q loss based on the observed reward vectors. MORL-FB achieves strong performance in a variety of discrete and continuous control tasks, offering superior efficiency, better generalization, and zero-shot cross-objective transfer. A key limitation, inherited from FB-based RFRL, is the need for more advanced exploration, especially in complex or sparse-reward environments, where dedicated strategies are crucial. Future work includes exploring RFRL methods beyond FB, such as representation learning of successor measures (Agarwal et al., 2025; Farebrother et al., 2023) and learning distance-preserving state representations (Park et al., 2024), to further reveal RFRL's advantages in MORL.

## ACKNOWLEDGMENT

This research is partially supported by the National Science and Technology Council (NSTC) of Taiwan under Grant Numbers 114-2628-E-A49-002 and 114-2634-F-A49-002-MBK. We also thank the National Center for High-performance Computing (NCHC) for providing computational and storage resources.

## ETHICS STATEMENT

Our work develops and evaluates reinforcement learning methods purely in simulated environments, without involving human subjects or sensitive data. This submission follows the code of ethics.

## REPRODUCIBILITY STATEMENT

We release our code in the supplementary material and describe the commands needed to execute the code in a Readme file attached in the supplementary material. Additionally, we attach the list of package dependencies which can be used to build the environment.

## THE USE OF LARGE LANGUAGE MODELS (LLMS)

Large language models (LLMs) were employed exclusively for language editing and polishing of the manuscript. They were not used for designing methods, conducting experiments, or analyzing results.

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

APPENDICES

## A  ADDITIONAL BACKGROUND: SUCCESSOR MEASURE AND FORWARD-BACKWARD REPRESENTATIONS

### A.1  SUCCESSOR MEASURE

Recall that in standard single-objective RL, the Q-function under a policy $\pi$ with respect to a reward function $R : \mathcal{S} \times \mathcal{A} \to \mathbb{R}$ is defined as

$$Q^\pi(s, a) := \mathbb{E}_\pi \left[ \sum_{t=0}^\infty \gamma^t R(s_t, a_t) \mid s_0 = s, a_0 = a \right], \tag{7}$$

which captures the long-term expected discounted reward under the policy $\pi$. One way to interpret the Q function is through the lens of *successor measure* $\mathcal{M}^\pi : \mathcal{S} \times \mathcal{A} \to \Delta(\mathcal{S} \times \mathcal{A})$, which reflects the discounted, expected future occupancy of the state-action pairs in $X$ when starting from $(s, a)$ and following policy $\pi$. Formally, the successor measure $\mathcal{M}^\pi$ is defined as follows: For any subset

$X \subset \mathcal{S} \times \mathcal{A}$, define

$$\mathcal{M}^\pi(s, a, X) := \mathbb{E}_\pi \left[ \sum_{t=0}^{\infty} \gamma^t \cdot \mathbb{I}\{(s_t, a_t) \in X\} \Big| s_0 = s, a_0 = a \right], \tag{8}$$

where $\mathbb{I}\{\cdot\}$ is the indicator function. The key property is that this measure $\mathcal{M}^\pi$ is agnostic to the reward function $R$ as it only depends on the environment dynamics and the policy $\pi$. Note that viewing $\mathcal{M}^\pi$ as a measure can deal with both the discrete and continuous cases. For ease of exposition, in the sequel, we focus on the case of discrete state and action spaces. With that said, we also let $\mathcal{M}^\pi(s, a, s', a')$ denote the successor measure of the state-action pair $(s', a')$ when starting from $(s, a)$ and following policy $\pi$. We can connect the Q function in Equation (7) with the successor measure in Equation (8) as follows:

$$Q^\pi(s, a) = \sum_{(s', a') \in \mathcal{S} \times \mathcal{A}} \mathcal{M}^\pi(s, a, s', a') \cdot R(s', a'). \tag{9}$$

Crucially, we can make two observations: (i) If we can directly learn the successor measure of a policy $\pi$, then we can derive the corresponding Q function in a zero-shot manner given any reward function. (ii) For any reward function $R$, if we can learn the successor measure of an optimal policy for $R$ (denoted by $\pi_R^*$), then we can learn the optimal Q function $Q^{\pi_R^*}$ and thereafter derive an optimal policy for $R$. Therefore, if we can directly encode the information about $\pi_R^*$ in the successor measure, then we can learn the optimal Q function for any possible reward function.

## A.2 FORWARD-BACKWARD (FB) REPRESENTATIONS

Our MORL-FB method is built on the Forward-Backward (FB) representation approach (Touati & Ollivier, 2021; Touati et al., 2023), which provides a compact, low-rank factorization of the successor measure. The FB framework decomposes the successor measure into the product of two vector-valued functions, namely a Forward function $\mathbf{F} : \mathcal{S} \times \mathcal{A} \times \mathbb{R}^{d_z} \to \mathbb{R}^{d_z}$ and a Backward function $\mathbf{B} : \mathcal{S} \times \mathcal{A} \to \mathbb{R}^{d_z}$. Let $\pi_z$ be a policy that depends on some vector $z \in \mathbb{R}^{d_z}$. Specifically, under the FB framework, the successor measure $\mathcal{M}^\pi$ is represented by:

$$\mathcal{M}^{\pi_z}(s, a, s', a') = \mathbf{F}(s, a, z)^\top \mathbf{B}(s', a'), \tag{10}$$

where $z$ acts as some latent vector that captures the required information about the policy $\pi_z$. Using this factorization, the Q-function for any policy $\pi_z$ can be expressed as

$$Q^{\pi_z}(s, a) = \sum_{(s', a') \in \mathcal{S} \times \mathcal{A}} \mathbf{F}(s, a, z)^\top \mathbf{B}(s', a') R(s', a'), \tag{11}$$

$$= \mathbf{F}(s, a, z)^\top \underbrace{\left( \sum_{(s', a') \in \mathcal{S} \times \mathcal{A}} \mathbf{B}(s', a') R(s', a') \right)}_{=: z_R}. \tag{12}$$

Note that Equation (12) holds for *any* $z$ (and hence any policy $\pi_z$). Therefore, we have the freedom to set $z_R$ to be the latent vector that results in an optimal policy for the reward function $R$, *i.e.*, $\pi_{z_R} \equiv \pi_R^*$. To achieve this, we shall simply enforce the following Bellman optimality equation (Touati & Ollivier, 2021), *i.e.*,

$$\pi_R^*(s) = \arg\max_{a \in \mathcal{A}} \mathbf{F}(s, a, z_R)^\top z_R. \tag{13}$$

The final optimal policy for the given preference is then directly constructed by maximizing the approximated Q-function: Therefore, if $\mathbf{F}$ and $\mathbf{B}$ are well learned, then one can directly retrieve an optimal policy for any reward function $R$ given at test time by computing $z_R$ and apply Equation (13). This factorization provides the foundation of our MORL-FB algorithm to efficiently explore and generalize across preferences during training.

---

**Algorithm 2** MORL-FB

---

1: **Input:** Network parameters $\theta, \bar{\theta}, \omega, \bar{\omega}, \eta, \bar{\eta}$, preference sampling distribution $\mathcal{P}_{\boldsymbol{\lambda}}$, preference set $\Lambda$, actor learning rates $\mu_\pi$, FB presentation learning rate $\mu_{\text{FB}}$, $\boldsymbol{z}$ dimension $d_z$, sample number $n_s$, update frequency $n_u$, warm up steps $n_w$, and target smoothing coefficient $\tau$
2: Initialize networks $\mathbf{F}_\theta, \mathbf{B}_\omega, \pi_\eta$ and target networks $\mathbf{F}_{\bar{\theta}}, \mathbf{B}_{\bar{\omega}}, \pi_{\bar{\eta}}$
3: Initialize replay buffer $\mathcal{M} \leftarrow \varnothing$
4: **for** each iteration $i$ **do**
5:     Sample a preference vector $\boldsymbol{\lambda} \sim \mathcal{P}_{\boldsymbol{\lambda}}$
6:     **if** $i \leq n_w$ **then**                                                  ▷ Warm-up stage
7:         Sample $\boldsymbol{z}$ from a multivariate normal distribution $\mathcal{N}(\mathbf{0}, \mathbb{I}^{d_z})$
8:         Normalize $\boldsymbol{z}$ such that $\boldsymbol{z} \leftarrow \sqrt{d_z}\frac{\boldsymbol{z}}{\|\boldsymbol{z}\|_2}$
9:     **else**
10:         $\boldsymbol{z} \leftarrow$ PG-Explore($\boldsymbol{\lambda}$)
11:     **end if**
12:     **for** each environment step $t$ **do**
13:         $a_t \sim \pi_\eta(\cdot|s_t; \boldsymbol{\lambda})$
14:         $s_{t+1} \sim \mathcal{P}(\cdot|s_t, a_t)$
15:         $\mathcal{M} \leftarrow \mathcal{M} \bigcup \{(s_t, a_t, \boldsymbol{r}_t, \boldsymbol{\lambda}_t, s_{t+1})\}$
16:     **end for**
17:     **for** each gradient step $j$ **do**
18:         Sample a batch of transitions $\{(s, a, \boldsymbol{r}, \boldsymbol{\lambda}, s')\}$ from the replay buffer $\mathcal{M}$
19:         $\boldsymbol{z}_j \leftarrow$ PG-Explore($\boldsymbol{\lambda}$)
20:         $\theta \leftarrow \theta - \mu_{\text{FB}}\nabla_\theta(\mathcal{L}_Q(\theta; \boldsymbol{\lambda}) + \mathcal{L}_{\text{M}}(\theta, \omega; \boldsymbol{\lambda}))$
21:         $\omega \leftarrow \omega - \mu_{\text{FB}}\nabla_\omega(\mathcal{L}_n(\omega; \boldsymbol{\lambda}) + \mathcal{L}_{\text{M}}(\theta, \omega; \boldsymbol{\lambda}))$
22:     **end for**
23:     **if** $i \% n_u == 0$ **then**
24:         $\eta \leftarrow \eta - \mu_\pi\nabla_\eta\mathcal{L}_\pi(\eta; \boldsymbol{\lambda})$
25:         $\bar{\theta} \leftarrow \tau\theta + (1-\tau)\bar{\theta}$
26:         $\bar{\omega} \leftarrow \tau\omega + (1-\tau)\bar{\omega}$
27:         $\bar{\eta} \leftarrow \tau\eta + (1-\tau)\bar{\eta}$
28:     **end if**
29: **end for**
30: **function** PG-EXPLORE($\boldsymbol{\lambda}$)
31:     Sample a batch $\mathcal{D}$ of $n_s$ non-terminal transitions $\{(s, a, \boldsymbol{r}, s')\}$ from $\mathcal{M}$
32:     $\boldsymbol{z} \leftarrow \sum_{(s,a,\boldsymbol{r},s') \in \mathcal{D}} \frac{\mathbf{B}_\omega(s,a)\boldsymbol{r}^\top\boldsymbol{\lambda}}{n_s}$
33:     Normalize $\boldsymbol{z}$ such that $\boldsymbol{z} \leftarrow \sqrt{d_z}\frac{\boldsymbol{z}}{\|\boldsymbol{z}\|_2}$
34:     **return** $\boldsymbol{z}$
35: **end function**

---

## B DETAILED PSEUDO CODE OF MORL-FB

Algorithm 2 details the proposed MORL-FB method. Initially, during the warm-up phase (lines 6-8), the latent vector $\boldsymbol{z}$ is sampled from a standard multivariate normal distribution. After the warm-up, $\boldsymbol{z}$ is determined using the preference-guided sampling scheme (line 10). This $\boldsymbol{z}$ is then used to generate trajectories within the environment, which are stored in the replay buffer $\mathcal{M}$ (lines 12-16). Model updates are performed by sampling transitions from $\mathcal{M}$ (lines 17-22). A delayed actor update mechanism is employed for the actor model (lines 23-24), and target networks are updated via a soft update scheme (lines 25-27). The Preference-Guided Exploration function (lines 30-35) normalizes the sampled latent vector $\boldsymbol{z}$ (line 33) as $\boldsymbol{z} \leftarrow \sqrt{d_z}\frac{\boldsymbol{z}}{\|\boldsymbol{z}\|_2}$. This normalization step, motivated by the prior work (Touati et al., 2023), has been observed to improve performance.

As shown in Algorithm 2, the training of MORL-FB involves the following loss functions:

**Measure Loss.** The Measure loss, $\mathcal{L}_{\text{M}}(\theta, \omega; \boldsymbol{z}_{\boldsymbol{\lambda}})$, is central to learning a task-agnostic representation of environment dynamics, $\mathbf{F}_\theta(s_t, a_t, \boldsymbol{z}_{\boldsymbol{\lambda}})$, encoding command-conditioned successor measures. It enforces Bellman consistency for these measures when projected onto a learned basis $\mathbf{B}_\omega(s', a')$, as shown in Equation (14). This mechanism, drawn from (Touati et al., 2023), aims to separate the

environment structure from specific rewards. This disentanglement is crucial for enabling zero-shot generalization, allowing the agent to understand "what happens next" irrespective of the immediate goal, forming a reusable foundation for various tasks.

$$
\begin{aligned}
\mathcal{L}_{\mathrm{M}}(\mathbf{F}_\theta, \mathbf{B}_\omega; \boldsymbol{z_\lambda}) =& \mathbb{E}_{\substack{(s_t, a_t, s_{t+1}) \sim \mathcal{D} \\ (s', a') \sim \mathcal{D}}} \big[ (\mathbf{F}_\theta(s_t, a_t, \boldsymbol{z_\lambda})^\top \mathbf{B}_\omega(s', a') \\
& - \gamma \mathbf{F}_{\bar{\theta}}(s_{t+1}, \pi(s_{t+1}, \boldsymbol{z_\lambda}), \boldsymbol{z_\lambda})^\top \mathbf{B}_{\bar{\omega}}(s', a'))^2 \big] \\
& - 2 \mathbb{E}_{(s_t, a_t, s_{t+1}) \sim \mathcal{D}} [\mathbf{F}_\theta(s_t, a_t, \boldsymbol{z_\lambda})^\top \mathbf{B}_\omega(s_{t+1}, a_{t+1})].
\end{aligned}
\tag{14}
$$

where $\rho$ denotes the underlying distributions of the dataset.

**Auxiliary Q Loss.** To ensure the learned representation $\mathbf{F}_\theta$ is relevant for decision-making, the Auxiliary Q Loss, $\mathcal{L}_Q(\theta; \boldsymbol{z_\lambda})$, connects it to task-specific values. When explicit reward signals $r_t$ and corresponding preferences $\boldsymbol{\lambda}$ are available, Equation (15) minimizes a standard temporal difference error. This is vital for MORL contexts, effectively teaching $\mathbf{F}_\theta$ to support optimizing diverse rewards.

$$
\mathcal{L}_Q(\theta; \boldsymbol{z_\lambda}) = \mathbb{E}_{(s_t, a_t, r_t, s_{t+1}) \sim \mathcal{D}} \left[ \left( \mathbf{F}_\theta(s_t, a_t, \boldsymbol{z_\lambda})^\top \boldsymbol{z_\lambda} - \left( \boldsymbol{\lambda}^\top \boldsymbol{r}_t + \gamma \mathbf{F}_{\bar{\theta}}(s_{t+1}, \pi_{\bar{\eta}}(s_{t+1}), \boldsymbol{z_\lambda})^\top \boldsymbol{z_\lambda} \right) \right)^2 \right].
\tag{15}
$$

**Orthonormality Loss.** The Orthonormality Regularization Loss, $\mathcal{L}_{\mathrm{n}}(\omega)$, acts as a crucial regularizer for the learned basis functions $\mathbf{B}_\omega(s, a)$. Its purpose, as reflected in Equation (16), is to promote a well-conditioned and non-degenerate basis. By encouraging properties such as orthogonality between basis vectors and unit norm, this loss helps prevent representational collapse and redundancy within $\mathbf{B}_\omega$. This, in turn, ensures that the successor measures are projected onto a stable and diverse set of features, enhancing the robustness and quality of the learned representations $\mathbf{F}_\theta$.

$$
\mathcal{L}_{\mathrm{n}}(\omega) = \mathbb{E}_{(s, a) \sim \mathcal{D}, (s', a') \sim \mathcal{D}} \left[ \left( \mathbf{B}_\omega(s, a)^\top \mathbf{B}_\omega(s', a') \right)^2 - \|\mathbf{B}_\omega(s, a)\|_2^2 - \|\mathbf{B}_\omega(s', a')\|_2^2 \right].
\tag{16}
$$

**Policy Loss.** The agent's behavior is refined through the Policy Optimization Loss, $\mathcal{L}_\pi(\eta; \boldsymbol{z_\lambda})$, which trains the policy $\pi_\eta$ within an actor-critic paradigm. The actor's objective is to maximize the Q-values estimated by the critic, where these Q-values are derived from the learned representation as $Q(s, a; \boldsymbol{z_\lambda}) = \mathbf{F}_\theta(s, a, \boldsymbol{z_\lambda})^\top \boldsymbol{z_\lambda}$ (Equation (17)). This loss drives the policy to select actions that are optimal for the task specified by the current command $\boldsymbol{z_\lambda}$. It thus enables the agent to translate its universal understanding of the environment into effective, task-adaptive behavior.

$$
\mathcal{L}_\pi(\eta; \boldsymbol{z_\lambda}) = \mathbb{E}_{s \sim \mathcal{D}} \left[ -Q(s, \pi_\eta(s); \boldsymbol{z_\lambda}) \right], \quad \text{where } Q(s, a; \boldsymbol{z_\lambda}) = \mathbf{F}(s, a, \boldsymbol{z_\lambda})^\top \boldsymbol{z_\lambda}.
\tag{17}
$$

## C  DETAILED CONFIGURATIONS OF EXPERIMENTS

In this section, we describe the experimental setup used to evaluate the performance of our approach. We detail the hyperparameters used in our experiments, as well as the reference points chosen for HV evaluation across different environments.

### C.1  EVALUATION ENVIRONMENTS

We evaluate the performance of our proposed method, MORL-FB, across a diverse set of multi-objective reinforcement learning environments. These environments, detailed below, encompass both established MuJoCo-based locomotion tasks and discrete problems, allowing us to assess its adaptability across distinct settings with varying state spaces, action spaces, and objective numbers.

**MuJoCo-Based Continuous Control:**

- **Halfcheetah2d:** The state space and action space are defined as $\mathcal{S} \subseteq \mathbb{R}^{17}$ and $\mathcal{A} \subseteq \mathbb{R}^6$, respectively. The two objectives for this environment are maximizing moving speed along the x-axis and minimizing energy cost.

- **Walker2d:** The state space and action space are defined as $\mathcal{S} \subseteq \mathbb{R}^{17}$ and $\mathcal{A} \subseteq \mathbb{R}^6$, respectively. The two objectives for this environment are maximizing moving speed along the x-axis and minimizing energy cost.

- **Hopper3d:** The state space and action space are defined as $\mathcal{S} \subseteq \mathbb{R}^{11}$ and $\mathcal{A} \subseteq \mathbb{R}^3$. The three objectives include maximizing moving speed along the x-axis, maximizing jumping height along the z-axis, and minimizing energy cost.

- **Ant3d:** The state space and action space are defined as $\mathcal{S} \subseteq \mathbb{R}^{27}$ and $\mathcal{A} \subseteq \mathbb{R}^8$. The three objectives are maximizing moving speed along the x-axis, maximizing moving speed along the y-axis, and minimizing energy cost.

- **Humanoid2d:** The state space and action space are defined as $\mathcal{S} \subseteq \mathbb{R}^{376}$ and $\mathcal{A} \subseteq \mathbb{R}^{17}$. The two objectives are maximizing moving speed along the x-axis and minimizing energy cost. Additionally, we set the healthy reward parameter to 1.0 to encourage exploration and stability.

- **Humanoid5d:** The state space and action space are defined as $\mathcal{S} \subseteq \mathbb{R}^{376}$ and $\mathcal{A} \subseteq \mathbb{R}^{17}$. This environment has five objectives: maximizing moving speed along the x-axis, maximizing moving speed along the y-axis, maximizing angular velocity on the left elbow, maximizing angular velocity on the right elbow, and minimizing energy cost. Similar to Humanoid2d, the healthy reward parameter is set to 1.0 to ensure meaningful evaluation.

**Classic Control:**

- **Deep Sea Treasure (DST):** The Deep Sea Treasure environment is a classic MORL problem in which the agent controls a submarine in a 2D grid world (Vamplew et al., 2011). The two conflicting objectives are typically maximizing collected treasure value and minimizing the time cost of collection.

- **Fruit Tree Navigation (FTN):** This environment is structured as a full binary tree of depth $d = 5, 6,$ or $7$ (Yang et al., 2019). The agent navigates from the root to a leaf node. Every leaf contains a fruit with values for six objectives: Protein, Carbs, Fats, Vitamins, Minerals, and Water.

### C.2 EXPERIMENTAL SETUP

To begin with, we describe the hyperparameters of the benchmark MORL methods and the proposed MORL-FB for better reproducibility.

**Hyperparameters for Experiments.** To ensure a fair comparison, for those benchmark methods that already provide tuned task-specific hyperparameters on MuJoCo, we primarily refer to their original papers for the hyperparameter configurations, including PGMORL (Xu et al., 2020) and CAPQL (Lu et al., 2023). Table 1 and Table 3 list the detailed hyperparameters used in our experiments. For PGMORL, the hyperparameters reflect its evolutionary population-based design. The parameter $n$ defines the number of parallel reinforcement learning tasks in each generation. Each task includes $m_w$ warm-up iterations and $m_t$ evolutionary iterations. $P_{\text{num}}$ and $P_{\text{size}}$ define the number and size of the performance buffers. The PPO parameters used across all environments are summarized in Table 2. For CAPQL, the hyperparameter $\alpha$ controls the strength of a concave regularization term added to the reward. The general hyperparameters shared by CAPQL are listed in Table 4.

Table 1: Hyperparameters of PGMORL.

| Environments | $n$ | $m_w$ | $m_t$ | $P_{\text{num}}$ | K | $P_{\text{size}}$ | $\alpha$ |
|---|---|---|---|---|---|---|---|
| HalfCheetah2d | 6 | 80 | 20 | 100 | 2 | 7 | $-1$ |
| Walker2d | 6 | 80 | 20 | 100 | 2 | 7 | $-1$ |
| Hopper3d | 15 | 200 | 40 | 210 | 2 | 7 | $-10^6$ |
| Ant3d | 15 | 200 | 40 | 210 | 2 | 7 | $-10^6$ |
| Humanoid2d | 6 | 200 | 40 | 100 | 2 | 7 | $-1$ |
| Humanoid5d | 35 | 200 | 40 | 550 | 2 | 7 | $-10^6$ |

Table 2: PPO hyperparameters used in PGMORL.

| Parameter | Value |
|---|---|
| Timesteps per actor batch | 2,048 |
| Processes number | 4 |
| Learning rate | $3 \times 10^{-4}$ |
| Discount factor ($\gamma$) | 0.995 |
| GAE lambda | 0.95 |
| Batch size | 32 |
| PPO epochs | 10 |
| Entropy coefficient | 0 |
| Value loss coefficient | 0.5 |

Table 3: Augmentation strength of CAPQL.

| Environments | $\alpha$ |
|---|---|
| HalfCheetah2d | 0.1 |
| Walker2d | 0.05 |
| Hopper3d | 0.2 |
| Ant3d | 0.2 |
| Humanoid2d | 0.005 |
| Humanoid5d | 0.005 |

Table 4: Hyperparameters of CAPQL.

| Parameter | Value |
|---|---|
| Optimizer | Adam |
| Learning rate | $3 \times 10^{-4}$ |
| Discount factor ($\gamma$) | 0.99 |
| Number of hidden units per layer | 256 |
| Replay buffer size | $10^6$ |
| Batch size | 256 |
| Nonlinearity | ReLU |
| Target smoothing coefficient ($\tau$) | 0.005 |

For algorithms without officially tuned or specified hyperparameters, we perform hyperparameter optimization (HPO) by following (Eimer et al., 2023). Specifically, we employ Bayesian optimization by using the Weights & Biases Sweeps. Each episode refers to a simulation run for 1M environment steps, and the optimization is performed over 10 episodes. Tables 5 to 11 summarize the search ranges and the final selected hyperparameters.

- **MORL-FB:** For MORL-FB, which is built on the implementation of the Twin Delayed Deep Deterministic Policy Gradient (TD3) algorithm (Fujimoto et al., 2018), we prioritized the tuning of parameters inherent to TD3. Specifically, this included critical elements such as learning rates for actor and critic networks, the target network update rate, and policy delay, which are known to significantly influence the stability and performance of TD3-based agents.

- **Q-Pensieve:** Our tuning efforts centered on parameters associated with its core Q-Snapshot mechanism (Hung et al., 2023). Given that Q-Pensieve's efficacy in improving sample efficiency stems from the storage and utilization of these Q-function snapshots, parameters governing the snapshot buffer, the frequency of snapshot capture, and their influence on the policy update were key areas of focus during hyperparameter optimization.

- **PD-MORL:** For PD-MORL, which can be viewed as the multi-objective extensdion of TD3 (Fujimoto et al., 2018), we prioritized the tuning of parameters inherent to TD3. This

included critical elements such as learning rates for actor and critic networks, the target network update rate, and policy delay, which are known to significantly influence the stability and performance of TD3-based agents.

- **MORL/D:** For MORL/D, which is built upon SAC, we tuned the standard hyperparameters of SAC. In addition, we focused on four components that are central to the MORL/D framework: population size, neighborhood size, scalarization method, and weight adaptation method. These elements directly affect how the algorithm decomposes the multi-objective space and maintains policy diversity.

- **PCN:** For PCN, since the hyperparameter settings for continuous control tasks such as MuJoCo environments were not specified in (Reymond et al., 2022), we tuned three critical parameters: learning rate, batch size, and hidden dimension, which affect training stability, sample efficiency, and the model's ability to generalize across diverse preference vectors.

- **SFOLS:** For SFOLS, since we utilize its official implementation for discrete problems and extend it with a TD3 backbone for evaluation on continuous control tasks, we tuned the parameters inherent to TD3, including learning rates for the actor and critic networks, policy noise, and target policy smoothing noise. These components are known to significantly influence the stability and performance of TD3-based agents.

- **GPI-PD:** For GPI-PD, since GPI-PD relies on a learned dynamics model for sample generation, we tuned four key parameters: dynamics rollout length, rollout frequency, model training frequency, and real data ratio. These factors critically affect model accuracy, stability of planning updates, and the overall effectiveness of Dyna-style training.

Table 5: Hyperparameter tuning for MORL-FB.

| Parameter | Value | Optimal Value |
|---|---|---|
| Learning rate | $[0.0001, 0.01]$ | 0.0001 |
| Policy update delay | $\{1, 2, 5, 10\}$ | 10 |
| Steps per update | $\{1, 2, 5, 10\}$ | 1 |
| Latent dimension ($z$ dimension) | $\{50, 150, 300\}$ | 300 |
| Exploration noise std. | $[0.1, 1.0]$ | 0.1 |
| Target policy smoothing noise std. | $[0.1, 1.0]$ | 0.2 |

Table 6: Hyperparameter tuning for Q-Pensieve.

| Parameter | Value | Optimal Value |
|---|---|---|
| Learning rate | $[0.0001, 0.01]$ | 0.0001 |
| Q replay buffer size | $\{1, 2, 4\}$ | 4 |
| Preference set size | $\{1, 2, 4\}$ | 4 |

Table 7: Hyperparameter tuning for PD-MORL.

| Parameter | Value | Optimal Value |
|---|---|---|
| Learning rate | $[0.0001, 0.01]$ | 0.0003 |
| Exploration noise std. | $[0.1, 1.0]$ | 0.15 |
| Target policy smoothing noise std. | $[0.1, 1.0]$ | 0.25 |
| Policy update delay | $\{1, 2, 5, 10\}$ | 10 |

**Hyperparameters of MORL-FB.** The hyperparameters for the MORL-FB experiments were chosen to ensure fair evaluation and stable learning, guided by prior research and the HPO results. The

Table 8: Hyperparameter tuning for PCN.

| Parameter | Value | Optimal Value |
|---|---|---|
| Learning rate | $[0.0001, 0.01]$ | $0.0023$ |
| Batch size | $\{64, 128, 256\}$ | $64$ |
| Number of hidden units per layer | $\{256, 512, 1024\}$ | $1,024$ |

Table 9: Hyperparameter tuning for MORL/D.

| Parameter | Value | Optimal Value |
|---|---|---|
| Learning rate | $[0.0001, 0.01]$ | $0.0013$ |
| Batch size | $\{64, 128, 256\}$ | $256$ |
| Number of hidden units per layer | $\{256, 512, 1024\}$ | $1,024$ |
| Pop size | $\{4, 6, 8\}$ | $6$ |
| Neighborhood size | $\{0, 1, 2\}$ | $1$ |
| Scalarization method | $\{none, ws\}$ | $ws$ |
| Weight adaptation method | $\{none, PSA\}$ | $PSA$ |

Table 10: Hyperparameter tuning for SFOLS.

| Parameter | Value | Optimal Value |
|---|---|---|
| Learning rate | $[0.0001, 0.01]$ | $0.0006$ |
| Batch size | $\{64, 128, 256\}$ | $256$ |
| Number of hidden units per layer | $\{256, 512, 1024\}$ | $1,024$ |
| Target smoothing coefficient $(\tau)$ | $[0.001, 0.02]$ | $0.0061$ |
| Exploration noise std. | $[0.1, 0.3]$ | $0.1736$ |
| Target policy smoothing noise std. | $[0.1, 0.5]$ | $0.4232$ |

Table 11: Hyperparameter tuning for GPI-PD.

| Parameter | Value | Optimal Value |
|---|---|---|
| Learning rate | $[0.0001, 0.01]$ | $0.0003$ |
| Batch size | $\{64, 128, 256\}$ | $256$ |
| Number of hidden units per layer | $\{256, 512, 1024\}$ | $1,024$ |
| Dynamics rollout length | $[5, 20]$ | $8$ |
| Dynamics rollout freqency | $[50, 500]$ | $312$ |
| Dynamics train freqency | $[100, 500]$ | $139$ |
| Dynamics real ratio | $[0.05, 0.2]$ | $0.0845$ |

hyperparameters tuned via HPO are presented in Table 5, while the remaining values were primarily drawing from the default settings of original FB and TD3 algorithms. The complete configuration is detailed in Table 12.

**Reference Points for HV Evaluation.** We compute the HV indicator using predefined reference points (Ref. Point) for each environment. These reference points serve as baselines to measure the

Table 12: Hyperparameter configuration for MORL-FB experiments.

| Parameter | Value |
|---|---|
| Total number of environment steps | $3 \times 10^6$ |
| Replay buffer size | $1 \times 10^6$ |
| Latent dimension ($z$ dimension) | 300 |
| Interface batch size | 5,120 |
| Number of hidden units per layer | 1,024 |
| Preprocessing feature dimension | 512 |
| Batch size | 1,024 |
| Target smoothing coefficient ($\tau$) | 0.01 |
| Discount factor ($\gamma$) | 0.99 |
| Learning rate | $1 \times 10^{-4}$ |
| Policy update delay | 10 |
| Steps per update | 1 |
| Exploration noise std. | 0.1 |
| Target policy smoothing noise std. | 0.2 |
| Clipping parameter | 0.5 |
| Value loss coefficient | 1 |

coverage of the Pareto front obtained during training. Table 13 provides the specific reference points used in different environments.

Table 13: Reference points used for HV calculation in different environments.

| Environment | Ref. Point |
|---|---|
| Deep Sea Treasure | (0, -50) |
| Fruit Tree Navigation | (0, 0, 0, 0, 0, 0) |
| Halfcheetah2d | (0, -8000) |
| Walker2d | (0, -8000) |
| Hopper3d | (0, 0, -8000) |
| Hopper4d | (0, 0, -8000, 0) |
| Ant3d | (0, 0, -8000) |
| Humanoid2d | (0, -8000) |
| Humanoid5d | (0, 0, 0, 0, -8000) |

## C.3 COMPUTE RESOURCES

All models were trained on a workstation featuring a single NVIDIA RTX 4090 GPU, an Intel Core i7-13700K CPU, and 64 GB of system memory.

# D  ADDITIONAL EXPERIMENTAL RESULTS

## D.1  VISUALIZATION OF $z$ DISTRIBUTION IN DIFFERENT ENVIRONMENTS

We analyze the learned latent representations of MORL-FB by visualizing the distribution of sampled z across various environments using t-SNE (Van der Maaten & Hinton, 2008). These visualizations offer insight into the method's ability to capture the underlying structure of the multi-objective tasks. Consistent with the observations on Humanoid2d (Figure 5), the visualizations for Walker2d, Hopper3d, and Ant3d (Figure 8, Figure 9, and Figure 10, respectively) demonstrate that preference-guided sampling yields distinct distributions compared to sampling from a standard normal distribution. These results further support the hypothesis that preference-guided sampling promotes the exploration of a more diverse set of latent representations, which may contribute to improved generalization and adaptation on various objectives.

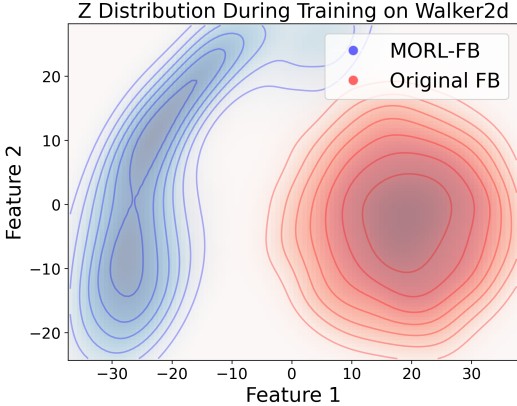

Figure 8: Empirical $z$ distribution under MORL-FB with preference-guided sampling versus Original FB with simple normal distributions on Walker2d.

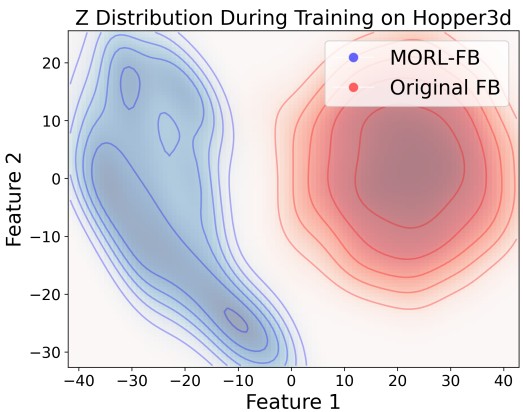

Figure 9: Empirical $z$ distribution under MORL-FB with preference-guided sampling versus Original FB with simple normal distributions on Hopper3d.

To shows the multi-modality of MORL-FB, we visualized the positions of latent variables $z$ inferred from different preferences on the t-SNE plot in Figure 11. This demonstrates that MORL-FB effectively encodes different preferences into separate regions of the latent space, leading to more diverse policies. To further illustrate this, we provide a demo of the policies learned by MORL-FB and vanilla FB in this link: https://imgur.com/a/ehx1v7q, where the z's are selected from different positions on the t-SNE plot.

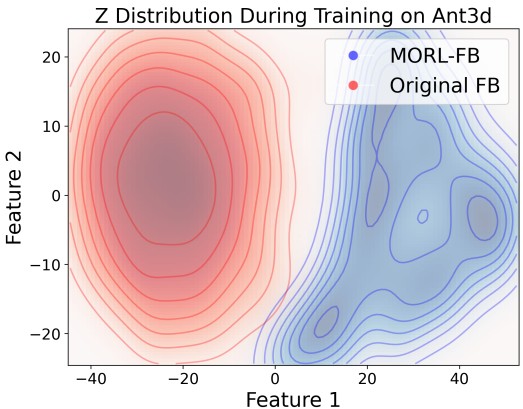

Figure 10: Empirical $z$ distribution under MORL-FB with preference-guided sampling versus Original FB with simple normal distributions on Ant3d.

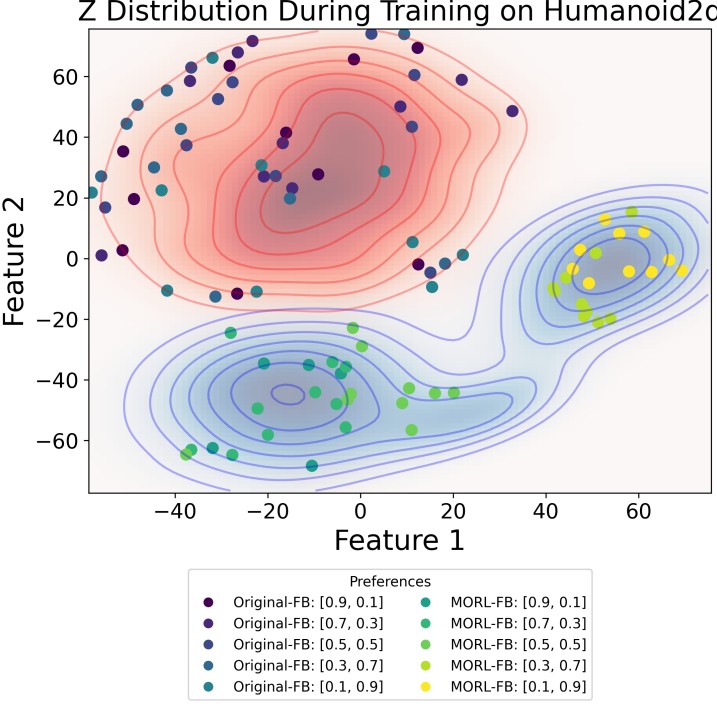

Figure 11: Empirical z distribution under MORL-FB with preference-guided sampling (blue) versus Original FB with simple normal distributions (red) on Humanoid2d. This figure is slightly different from Figure 5 due to the additional preference points and the inherent randomness of t-SNE.

## D.2 ABLATION STUDY

### D.2.1 EXPERIMENT ON PREFERENCE-GUIDED EXPLORATION

To assess the data efficiency of our proposed MORL-FB, we compared its performance against PD-MORL (Basaklar et al., 2023). PD-MORL necessitates three to five times more training samples to train its preference interpolator. Can MORL-FB maintain superior performance while reducing data requirements?

Table 14 reveals that MORL-FB consistently outperforms PD-MORL across various environments. Specifically, MORL-FB achieves higher UT values in five out of six tasks, demonstrating its effec-

tiveness in most scenarios. In more complex settings, such as MO-Humanoid, how does MORL-FB compare in HV results? The results indicate that MORL-FB remains competitive, underscoring its significant data efficiency gains. By eliminating the need for additional data to pretrain an interpolator, MORL-FB achieves competitive or superior performance while requiring significantly fewer training samples in multi-objective environments.

Table 14: Performance comparison between MORL-FB and PD-MORL (with interpolator) across key metrics (UT, HV, and ED) on various multi-objective tasks.

| Environments | Metrics | PD-MORL (w/i interpolator) | MORL-FB |
|---|---|---|---|
| Halfcheetah2d | UT($\times 10^3$) | $5.62 \pm 0.05$ | $\mathbf{7.69 \pm 0.08}$ |
| | HV($\times 10^8$) | $1.08 \pm 0.00$ | $\mathbf{1.24 \pm 0.00}$ |
| | ED | $0.07 \pm 0.01$ | - |
| Walker2d | UT($\times 10^3$) | $2.18 \pm 0.02$ | $\mathbf{2.23 \pm 0.03}$ |
| | HV($\times 10^7$) | $\mathbf{5.45 \pm 0.01}$ | $4.32 \pm 0.02$ |
| | ED | $0.56 \pm 0.00$ | - |
| Hopper3d | UT($\times 10^3$) | $2.26 \pm 0.01$ | $\mathbf{2.36 \pm 0.01}$ |
| | HV($\times 10^{11}$) | $\mathbf{1.08 \pm 0.00}$ | $1.15 \pm 0.00$ |
| | ED | $0.20 \pm 0.01$ | - |
| Ant3d | UT($\times 10^3$) | $\mathbf{3.59 \pm 0.06}$ | $3.43 \pm 0.22$ |
| | HV($\times 10^{11}$) | $\mathbf{4.20 \pm 0.04}$ | $4.18 \pm 0.04$ |
| | ED | $0.60 \pm 0.00$ | - |
| Humanoid2d | UT($\times 10^2$) | $2.93 \pm 0.07$ | $\mathbf{8.13 \pm 0.01}$ |
| | HV($\times 10^7$) | $1.06 \pm 0.01$ | $\mathbf{1.75 \pm 0.02}$ |
| | ED | $0.33 \pm 0.01$ | - |
| Humanoid5d | UT($\times 10^3$) | $0.93 \pm 0.04$ | $\mathbf{1.11 \pm 0.00}$ |
| | HV($\times 10^{15}$) | $6.64 \pm 0.09$ | $\mathbf{6.99 \pm 0.06}$ |
| | ED | $0.43 \pm 0.01$ | - |

### D.2.2 EXPERIMENTS ON $z$ DIMENSION

We investigate the impact of the $z$ dimension in the Hopper3d environment. As shown in Table 15, the performance metric increases with the $z$ dimension. However, when the $z$ dimension reaches 300, performance declines, likely due to insufficient training steps for the larger network.

Table 15: Empirical study on $z$ dimension on Hopper3d.

| $z$ dimension | Metrics | MORL-FB |
|---|---|---|
| 50 | UT($\times 10^3$) | $2.21 \pm 0.01$ |
| | HV($\times 10^{11}$) | $1.03 \pm 0.05$ |
| 100 | UT($\times 10^3$) | $2.25 \pm 0.00$ |
| | HV($\times 10^{11}$) | $1.13 \pm 0.01$ |
| 150 | UT($\times 10^3$) | $2.36 \pm 0.01$ |
| | HV($\times 10^{11}$) | $1.15 \pm 0.00$ |
| 300 | UT($\times 10^3$) | $2.01 \pm 0.01$ |
| | HV($\times 10^{11}$) | $0.97 \pm 0.02$ |

### D.2.3 EXPERIMENTS ON AUXILIARY LOSS

While Touati et al. (2023) also includes a similar auxiliary loss, there is one salient difference between theirs and our Q loss Equation (6): As the original FB is designed for RFRL, it does not have the reward signal available at training and hence needs to construct a pseudo reward as

$B^\top \mathbb{E}[BB^\top]^{-1}z$ (cf. Equation (9) in (Touati et al., 2023)). On the other hand, as MORL-FB addresses MORL and can observe vector reward signals at training, we propose to use the actual scalarized reward $\lambda^\top r$ in the Q loss. While this algorithmic difference appears seemingly subtle, this design makes a huge difference in the performance. Below we show an ablation study that compares MORL-FB with our Q loss (denoted as "Original") and MORL-FB with the auxiliary loss using pseudo reward in FB (denoted as "Pseudo Q Loss"). The results are summarized in Table 16 and Fig. 12. This shows that the auxiliary loss of FB cannot be directly applied and needs to be adapted properly in the context of MORL. Note that we use the term "auxiliary" since the original FB is directly built on the measure loss and hence the Q loss is auxiliary for learning FB representation, rather than being unimportant for MORL.

Table 16: Ablation study of MORL-FB on preference-guided exploration.

| Environments | Metrics | MORL-FB (w/o PG-Explore) | MORL-FB (Pseudo Q Loss) | MORL-FB (Ours) |
|---|---|---|---|---|
| Ant3d | UT($\times 10^3$) | $1.45 \pm 0.02$ | $1.27 \pm 0.16$ | $\mathbf{3.93 \pm 0.04}$ |
| | HV($\times 10^{11}$) | $1.25 \pm 0.02$ | $1.91 \pm 0.02$ | $\mathbf{3.85 \pm 0.01}$ |

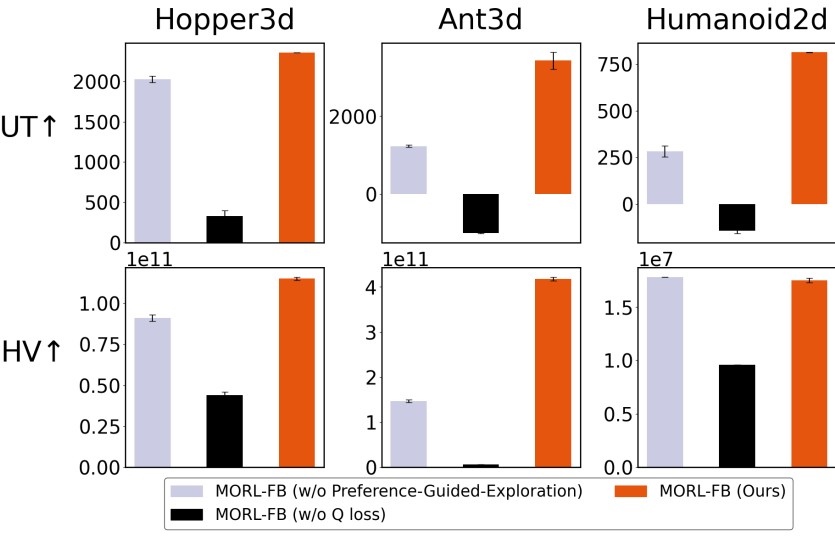

Figure 12: Evaluation of MORL-FB and its ablated versions across different environments. The results highlight the importance of preference-guided sampling and the auxiliary Q-loss for MORL performance.

### D.2.4 EXPERIMENTS ON MEASURE LOSS

Q-loss is surely important in our method as it guides our representation to learn the reward function used in the environment. On the other hand, both the preference-guided exploration and the measure loss are also essential to the success of MORL-FB based on Figure 12 and an additional ablation study on the measure loss shown below.

Table 17: Ablation study of MORL-FB on measure loss and Q loss.

| Environments | Metrics | MORL-FB (w/o measure loss) | MORL-FB (w/o Q loss) | MORL-FB (Ours) |
|---|---|---|---|---|
| Ant3d | UT($\times 10^3$) | $1.37 \pm 0.28$ | $-1.53 \pm 0.00$ | $\mathbf{3.93 \pm 0.04}$ |
| | HV($\times 10^{11}$) | $2.74 \pm 0.02$ | $0.00 \pm 0.00$ | $\mathbf{3.85 \pm 0.01}$ |

### D.2.5 Experiments on Q-loss Coefficient

In our implementation, the measure loss $\mathcal{L}_M$ and the Q loss $\mathcal{L}_Q$ are directly added together without an additional weighting term. We chose this default configuration because it already yields strong and stable performance across all evaluated tasks, and we did not observe a need for further hyperparameter tuning during development. Moreover, we have conducted an additional sensitivity analysis to better understand how the balance between these two losses affects performance. Specifically, we varied the Q-loss coefficient $\alpha_Q$ (with $\alpha_Q = 1$ as the default setting) and reported the corresponding results in Ant3d. Table 18 shows that MORL-FB's performance is largely insensitive to the choice of $\alpha_Q$, indicating that directly adding $\mathcal{L}_M$ and $\mathcal{L}_Q$ is a reasonable and robust design choice.

Table 18: Performance of MORL-FB with different Q-loss coefficients.

| Environments | Metrics | MORL-FB | | | |
| --- | --- | --- | --- | --- | --- |
| | | $\alpha_Q = 0.25$ | $\alpha_Q = 0.5$ | $\alpha_Q = 1$ (Default) | $\alpha_Q = 2$ |
| Ant3d | UT($\times 10^3$) | $3.76 \pm 0.41$ | $3.79 \pm 0.32$ | $3.77 \pm 0.11$ | $\mathbf{3.95 \pm 0.13}$ |
| | HV($\times 10^{11}$) | $4.03 \pm 0.22$ | $4.13 \pm 0.19$ | $3.95 \pm 0.13$ | $\mathbf{4.23 \pm 0.04}$ |

### D.3 Performance Comparison of MORL-FB Under Different Q-loss Coefficients

We evaluate MORL-FB and several baselines on two classic discrete control tasks: Deep Sea Treasure (DST) and Fruit Tree Navigation (FTN). DST features the fundamental trade-off between two conflicting objectives. FTN, on the other hand, involves a tree search process with 6-dimensional terminal rewards, as previously detailed in the evaluation environments section. The detailed configurations for these experiments are provided in Section C.

Figure 13 shows the performance of all the methods in UT, HV, and ED for discrete control tasks. Regarding ED, for each baseline algorithm ALG, we report ED(ALG, MORL-FB) to show the pairwise comparison. We can observate that MORL-FB consistently achieves competitive or superior performance across all three metrics on the discrete control tasks Deep Sea Treasure and Fruit Tree Navigation.

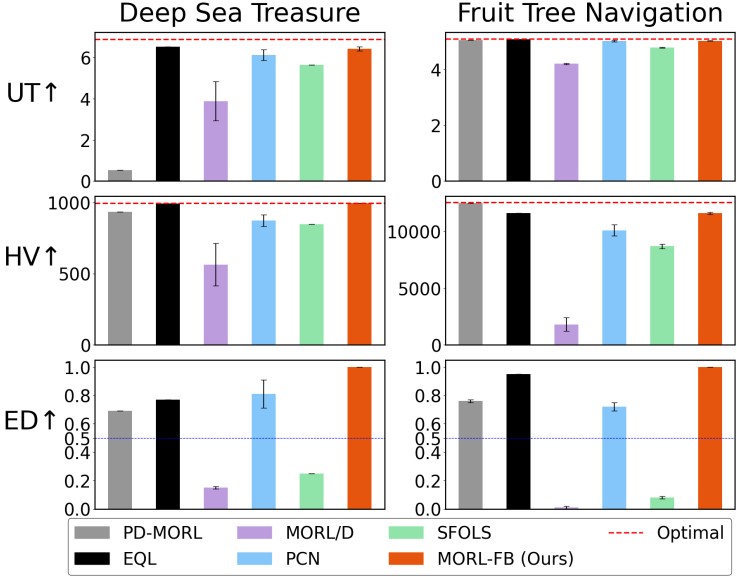

Figure 13: **Competitive Results of MORL-FB on Discrete Control Tasks.** We evaluate MORL-FB and several benchmark MORL algorithms on classic discrete control tasks in MO-Gymnasium. Performance is measured using UT, HV and ED. MORL-FB demonstrates competitive results against specialized discrete MORL algorithms.

### D.4 PERFORMANCE COMPARISON OF MORL-FB UNDER STATE-ACTION-BASED REWARDS AND STATE-BASED REWARDS

In this paper, we primarily focus on state-based rewards. However, as the original FB supports both state-based (Touati et al., 2023) and state-action-based rewards (Touati & Ollivier, 2021), MORL-FB can also be extended to state-action-based rewards by replacing $B(s)$ with $B(s, a)$. Since some MO MuJoCo rewards depend on both states and actions, we compare MORL-FB and the extended one. As shown in the Figure 14, the state-action-based variant yields slight performance improvements on several tasks.

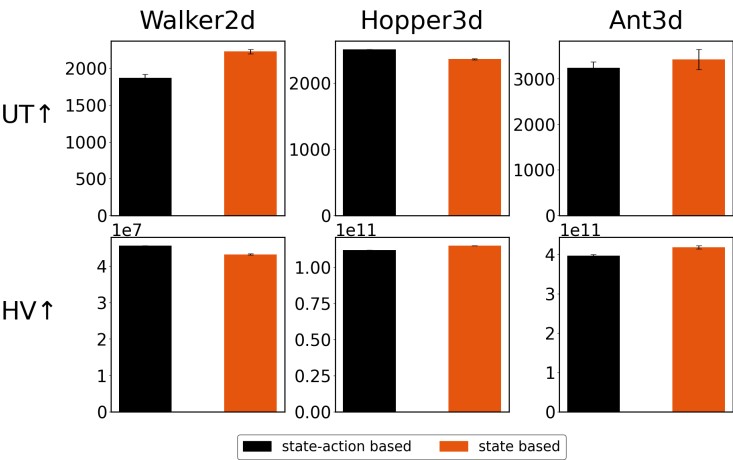

Figure 14: **Evaluation of MORL-FB with different reward function representations.** This figure presents the performance of MORL-FB when reward functions depend on states only (*i.e.,* $\boldsymbol{R}(s)$) versus state-action pairs (*i.e.,* $\boldsymbol{R}(s, a)$).

### D.5 PERFORMANCE COMPARISON OF MORL-FB UNDER STOCHASTIC REWARDS

As vanilla FB naturally handles stochastic rewards, MORL-FB inherits this capability. To further demonstrate this, we evaluated MORL-FB under stochastic rewards by adding zero-mean Gaussian noise $\mathcal{N}(0, \sigma^2)$, similar to prior work (Romoff et al., 2018; Hu et al., 2022). The result is shown in Figure 15.

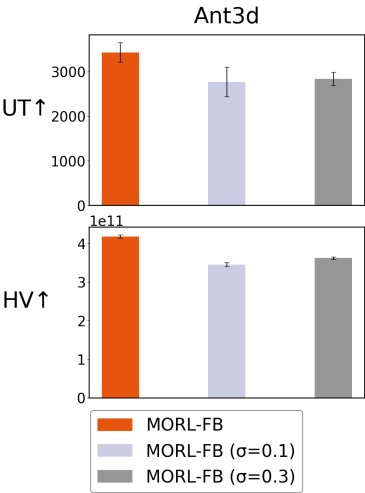

Figure 15: **Evaluation of MORL-FB under stochastic reward.** This figure assesses the performance of MORL-FB in environments featuring stochastic reward functions.

### D.6 PERFORMANCE COMPARISON OF MORL-FB WITH NONLINEAR SCALARIZATION

While we focus on linear scalarization in this paper, MORL-FB can be readily extended to nonlinear scalarization schemes by replacing $r^\top \lambda$ with $f_\lambda(r)$ when sampling $z$ for preference-guided exploration and when computing Q loss, where $f_\lambda(r)$ is the general scalarization function. This is feasible since the original FB is designed to handle any scalar reward function, and MORL-FB inherits this property from FB and can also handle nonlinear scalarization. To demonstrate this generalizability, we further evaluate MORL-FB on Halfcheetah2d by training under smooth Tchebycheff scalarization as $f_\lambda(r) = \mu \log\left(\sum_{i=1}^m \exp\left(\frac{\lambda_i(r - r_{ref})}{\mu}\right)\right)$, where $\lambda_i$ = is the $i$-th entry of preference vector, $\mu$ is the smoothing parameter and set to 0.1, and $r_{ref}$ is set to [2.0, 0.0] across training (Lin et al., 2024; Qiu et al., 2024). We see that MORL-FB still achieves comparably strong performance in HV and UT under nonlinear scalarization.

Table 19: Performance of MORL-FB with Smooth Tchebycheff scalarization.

| Environments | Metrics | MORL-FB (linear scalarization) | MORL-FB (Smooth Tchebycheff scalarization) |
|---|---|---|---|
| HalfCheetah2d | UT($\times 10^3$) | $7.69 \pm 0.08$ | $6.33 \pm 0.02$ |
| | HV($\times 10^8$) | $1.24 \pm 0.00$ | $1.00 \pm 0.01$ |

### D.7 SAMPLE EFFICIENCY OF MORL-FB

We demonstrate the sample efficiency of MORL-FB by evaluating its performance at an intermediate stage of 1.5M training steps. Notably, as shown in Figure 16, MORL-FB achieves superior HV and UT scores across most tasks compared to baseline methods trained for a full 3M steps. This indicates that MORL-FB can attain high performance with significantly fewer environment interactions. Further evidence, presented in Figure 17 and Figure 18, corroborates that MORL-FB reaches proficient performance levels with reduced training data.

### D.8 CROSS-OBJECTIVE TRANSFER CAPABILITY OF MORL-FB

To investigate how well MORL-FB handles transfer across different numbers of objectives, we conducted an empirical study on Hopper across different objective dimensions. We analyze the following cases:

- Hopper2d: Moving forward speed on the x-axis, control cost of the action
- Hopper3d: Moving forward speed on the x-axis, jumping height on the z-axis, control cost of the action
- Hopper4d: Moving forward speed on the x-axis, jumping height on the z-axis, jumping up speed on the z-axis, control cost of the action

Table 20 summarizes the quantitative results presented visually in Figure 7. MORL-FB consistently outperforms FB across all configurations in terms of utility and hypervolume.

### D.9 RFRL AS A SOURCE OF AUXILIARY TASKS

During training, the $z$ vectors computed for each preference $\lambda$ are diverse, covering both CCS and non-CCS policies. Learning from non-CCS policies serves as auxiliary tasks. From Figure 19, we find that the return vectors achieved by those $z$-induced polices at the 1.5 million training step span both non-CCS and CCS regions.

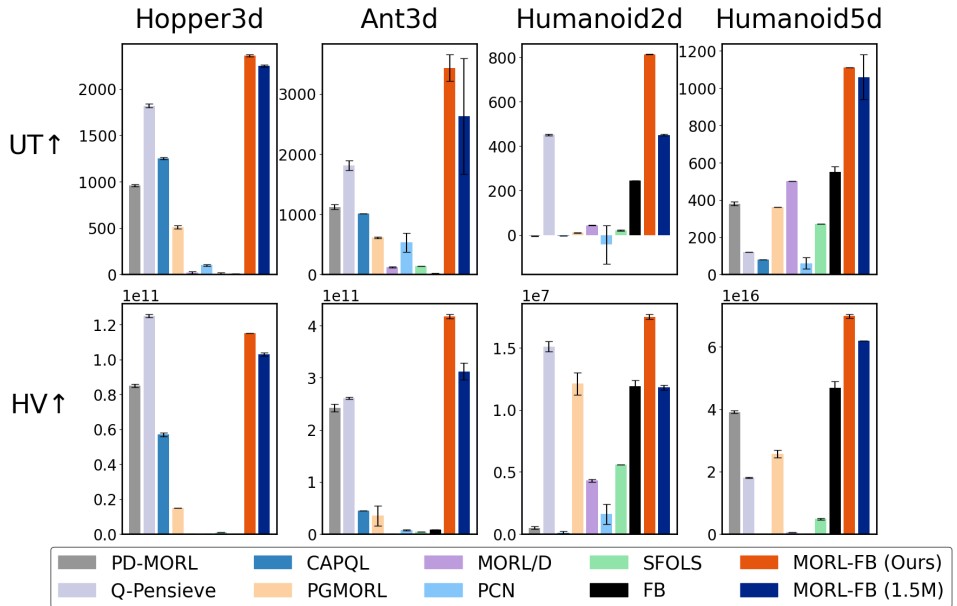

Figure 16: **Performance of MORL-FB on continuous control tasks.** We evaluate MORL-FB (1.5M training steps) against several benchmark MORL algorithms (3M training steps) on diverse continuous control tasks from MO-Gymnasium. Utilizing key metrics, these results demonstrate that MORL-FB outperforms baselines, achieving superior HV and UT across most tasks despite significantly fewer training steps.

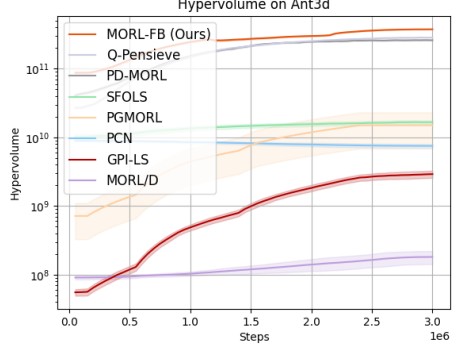

Figure 17: **Learning curves for MORL-FB and benchmark algorithms on Ant3d.** This figure presents the learning curves in terms of Hypervolume (HV) for MORL-FB and several benchmark MORL algorithms evaluated on Ant3d.

Figure 18: **Learning curves for MORL-FB and benchmark algorithms on Ant3d.** This figure presents the learning curves in terms of Utility (UT) for MORL-FB and several benchmark MORL algorithms evaluated on Ant3d.

## D.10    COMPARISON OF PARETO FRONTS

To evaluate the sample efficiency of MORL-FB, we conduct experiments on 2-objective MuJoCo tasks with a whole range of 21 preference vectors ([0.0, 1.0], [0.05, 0.95], [0.1, 0.9], ···, [1.0, 0.0]). As a comparison baseline, we consider SORL which trains a separate policy for each individual preference. Each single-object SAC (SOSAC) model requires 3M steps, resulting in a total training budget of 63M steps for all 21 preferences. By contrast, MORL-FB only uses 3M steps in total to learn policies that generalize across the entire preference set. Figure 20 shows the return vectors attained by MORL-FB and the collection of 21 SOSAC models on the Walker2d task. MORL-FB

Table 20: **Zero-shot cross-objective transfer from Hopper2d to Hopper3d and Hopper4d using vanilla FB and MORL-FB:** This figure presents the results demonstrating effective transfer by MORL-FB, supporting the efficacy of its proposed enhancements.

| Environments | Metrics | FB | MORL-FB |
|---|---|---|---|
| 2D to 3D | UT($\times 10^3$) | $0.02 \pm 0.00$ | $\mathbf{1.77 \pm 0.00}$ |
| | HV($\times 10^{10}$) | $0.40 \pm 0.00$ | $\mathbf{7.81 \pm 0.06}$ |
| | ED | $0.07 \pm 0.03$ | - |
| 2D to 4D | UT($\times 10^3$) | $0.06 \pm 0.00$ | $\mathbf{1.59 \pm 0.00}$ |
| | HV($\times 10^{13}$) | $0.05 \pm 0.00$ | $\mathbf{7.78 \pm 0.10}$ |
| | ED | $0.05 \pm 0.01$ | - |

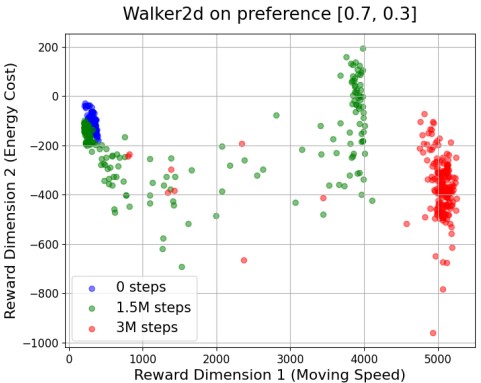

Figure 19: Return vectors (Moving Speed vs. Energy Cost) achieved at the initial, intermediate, and final training stages in Walker2d with preference [0.7, 0.3]: The scatter plot, particularly at 1.5M steps, highlights the diverse policies beyond the CCS policy, supporting that RFRL serves as auxiliary tasks beneficial for MORL.

achieves comparable or even superior return vectors with only 1/21 of the samples, demonstrating its strong sample efficiency and generalization ability across diverse preference vectors.

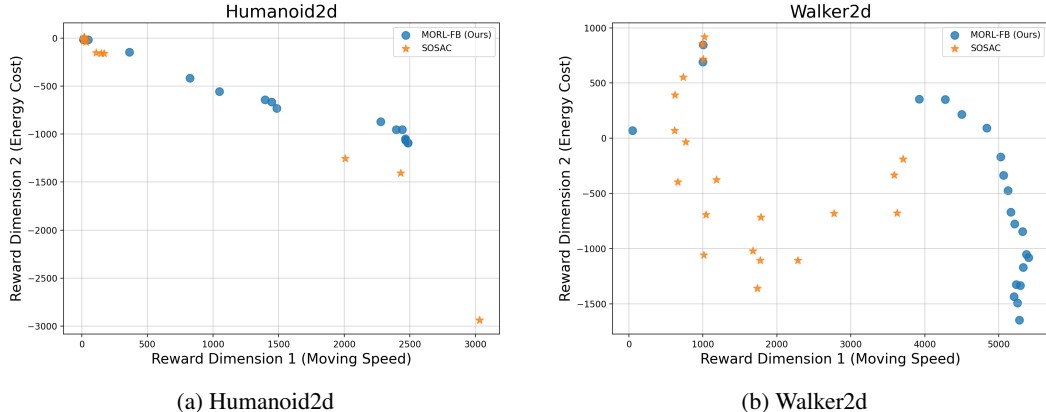

(a) Humanoid2d

(b) Walker2d

Figure 20: Return vectors (Moving Speed vs. Energy Cost) achieved under 21 different preference vectors $[0, 1], [0.05, 0.95], \ldots, [0.95, 0.05], [1, 0]$ across different methods. Each scatter cloud corresponds to the learned policies under a specific preference, illustrating how MORL methods adapt to diverse trade-offs between objectives.

### D.11 Robustness under Worst-Case Preferences

Beyond average performance metrics, it is crucial to assess the robustness of multi-objective reinforcement learning (MORL) algorithms under unfavorable preference settings. In practice, policies deployed in dynamic environments may encounter user preferences that significantly diverge from those seen during training. To capture this aspect, we evaluate each method under *worst-case preferences* using the Conditional Value-at-Risk (CVaR). Following the risk-sensitive reinforcement learning formulation of Tamar et al. (2015), we compute CVaR@0.1 as the mean of the lowest $10\%$ scalarized returns. Specifically, for each algorithm we uniformly sample $500$ preference vectors, apply linear scalarization to obtain utility values, sort the results, and average the worst $10\%$. This quantifies the expected return conditioned on being in the lowest $\alpha$-quantile:

$$\phi(\theta) = \mathbb{E}^{\theta}[R \mid R \leq \nu_{\alpha}(\theta)], \tag{18}$$

Table 21 reports CVaR@0.1 across HalfCheetah2d and Walker2d. MORL-FB achieves the highest CVaR in both environments, significantly outperforming prior baselines. These results indicate that MORL-FB not only excels on mean metrics but also demonstrates superior robustness against adverse preferences, avoiding catastrophic failures more effectively than existing methods.

Table 21: CVaR@0.1 performance across HalfCheetah2d and Walker2d. Higher is better.

| Algorithm | HalfCheetah2d | Walker2d |
|---|---|---|
| PD-MORL | 2425.91 | 284.89 |
| Q-Pensieve | 6275.89 | 1516.20 |
| CAPQL | 4874.64 | 1297.40 |
| PGMORL | 2630.02 | 1113.49 |
| MORL/D | 407.66 | 40.47 |
| PCN | -0.11 | 98.28 |
| SFOLS | 1390.73 | 402.19 |
| GPI-LS | 3884.67 | 1546.96 |
| GPI-PD | 3999.36 | 276.74 |
| MORL-FB (Ours) | **7123.87** | **2304.02** |

### D.12 Computational Cost Analysis

To provide a fair and comprehensive comparison of computational efficiency, we evaluated all methods under the same hardware environment. Each algorithm was trained for 100K environment steps on a workstation equipped with a single NVIDIA RTX 4090 GPU, an Intel Core i7-13700K CPU, and 64 GB of system memory. Table 22 reports the total wall-clock time required by each method, ensuring a fair comparison of time and resource usage across methods. Compared to strong baselines such as PD-MORL, Q-Pensieve, and GPI-LS, FB-MORL maintains a reasonable training time. While some methods require higher computational costs, our approach strikes a good balance between performance and efficiency.

### D.13 Detailed Experimental Results of Section 4

In this part, we will provide the table we use on plotting the bar chart on Section 4.

The results in Tables 23 to 26 correspond to the visualizations shown in Figure 2, Figure 4, Figure 12 and Figure 13, respectively.

Table 22: Wall-clock time comparison (100K steps).

| Algorithm | Wall-Clock Time (seconds) |
|---|---|
| PD-MORL | 1166 |
| CAPQL | 3369 |
| MORL/D | 793 |
| SFOLS | 550 |
| Q-Pensieve | 12960 |
| PGMORL | 4237 |
| PCN | 4445 |
| GPI-LS | 3611 |
| GPI-PD | 5237 |
| **MORL-FB (Ours)** | **1874** |

Table 23: Performance of MORL-FB and benchmark algorithms in discrete MORL environments.

| Environments | Metrics | PD-MORL (w/i interpolator) | Envelope | PCN | MORL/D | SFOLS | MORL-FB (Ours) | Optimal |
|---|---|---|---|---|---|---|---|---|
| Deep Sea Treasure | UT | $0.52 \pm 0.00$ | $6.53 \pm 0.00$ | $6.12 \pm 0.95$ | $3.88 \pm 0.26$ | $5.65 \pm 0.95$ | $6.43 \pm 0.10$ | 6.89 |
| | HV($\times 10^2$) | $9.33 \pm 0.00$ | $9.91 \pm 0.00$ | $8.71 \pm 1.49$ | $5.63 \pm 0.41$ | $8.46 \pm 0.01$ | $9.92 \pm 0.00$ | 9.92 |
| | ED | $0.69 \pm 0.00$ | $0.77 \pm 0.00$ | $0.81 \pm 0.01$ | $0.15 \pm 0.10$ | $0.25 \pm 0.00$ | - | - |
| Fruit Tree Navigation | UT | $5.03 \pm 0.00$ | $5.07 \pm 0.00$ | $5.00 \pm 0.02$ | $4.19 \pm 0.03$ | $4.77 \pm 0.01$ | $5.01 \pm 0.01$ | 5.08 |
| | HV($\times 10^4$) | $1.25 \pm 0.00$ | $1.16 \pm 0.00$ | $1.01 \pm 0.06$ | $0.18 \pm 0.05$ | $0.87 \pm 0.01$ | $1.16 \pm 0.01$ | 1.25 |
| | ED | $0.76 \pm 0.01$ | $0.95 \pm 0.00$ | $0.72 \pm 0.03$ | $0.01 \pm 0.01$ | $0.08 \pm 0.01$ | - | - |

Table 24: Performance of MORL-FB, PD-MORL, and Q-Pensieve under constrained preference training.

| Environments | Metrics | PD-MORL | Q-Pensieve | MORL-FB |
|---|---|---|---|---|
| Hopper3d | UT($\times 10^3$) | $1.05 \pm 0.02$ | $1.72 \pm 0.01$ | $\mathbf{2.26 \pm 0.01}$ |
| | HV($\times 10^{11}$) | $0.61 \pm 0.01$ | $0.88 \pm 0.01$ | $\mathbf{1.16 \pm 0.01}$ |
| | ED | $0.01 \pm 0.00$ | $0.11 \pm 0.00$ | - |
| Ant3d | UT($\times 10^3$) | $1.69 \pm 0.04$ | $0.49 \pm 0.02$ | $\mathbf{3.11 \pm 0.24}$ |
| | HV($\times 10^{11}$) | $2.18 \pm 0.03$ | $0.51 \pm 0.00$ | $\mathbf{3.17 \pm 0.05}$ |
| | ED | $0.22 \pm 0.02$ | $0.26 \pm 0.04$ | - |
| Humanoid2d | UT($\times 10^2$) | $-0.04 \pm 0.00$ | $4.51 \pm 0.38$ | $\mathbf{8.19 \pm 0.03}$ |
| | HV($\times 10^7$) | $0.06 \pm 0.00$ | $1.51 \pm 0.04$ | $\mathbf{1.83 \pm 0.01}$ |
| | ED | $0.00 \pm 0.00$ | $0.40 \pm 0.05$ | - |

Table 25: **Performance comparison of MORL-FB and its ablated versions across environments.** This table evaluates MORL-FB against variants lacking preference-guided exploration or the Q-loss component, showing their performance across different environments.

| Environments | Metrics | MORL-FB (w/o preference-guided exploration) | MORL-FB (w/o q loss) | MORL-FB |
|---|---|---|---|---|
| Hopper3d | UT($\times 10^3$) | $2.03 \pm 0.42$ | $0.33 \pm 0.07$ | $\mathbf{2.36 \pm 0.00}$ |
| | HV($\times 10^{11}$) | $0.91 \pm 0.02$ | $0.44 \pm 0.02$ | $\mathbf{1.15 \pm 0.01}$ |
| Ant3d | UT($\times 10^3$) | $1.23 \pm 0.03$ | $-1.01 \pm 0.01$ | $\mathbf{3.43 \pm 0.22}$ |
| | HV($\times 10^{11}$) | $1.47 \pm 0.03$ | $0.06 \pm 0.00$ | $\mathbf{4.18 \pm 0.04}$ |
| Humanoid2d | UT($\times 10^2$) | $2.38 \pm 0.30$ | $-1.43 \pm 0.14$ | $\mathbf{8.13 \pm 0.01}$ |
| | HV($\times 10^7$) | $\mathbf{1.78 \pm 0.00}$ | $0.96 \pm 0.00$ | $1.75 \pm 0.02$ |

Table 26: Comparative performance of MORL-FB and various benchmark algorithms across continuous control tasks in MuJoCo.

| Environments | Metrics | PD-MORL (w/o interpolator) | Q-Pensieve | CAPQL | PGMORL | MORL/D | PCN | SFOLS | GPI-LS | GPI-PD | FB | MORL-FB (Ours) |
|---|---|---|---|---|---|---|---|---|---|---|---|---|
| Halfcheetah2d | UT($\times10^3$) | $3.17\pm0.04$ | $6.85\pm0.01$ | $5.39\pm0.01$ | $2.72\pm0.03$ | $0.41\pm0.01$ | $0.00\pm0.00$ | $1.39\pm0.03$ | $4.15\pm0.84$ | $4.06\pm0.03$ | $1.26\pm0.00$ | $\mathbf{7.69\pm0.08}$ |
| | HV($\times10^8$) | $0.95\pm0.00$ | $1.13\pm0.00$ | $0.89\pm0.00$ | $0.73\pm0.01$ | $0.08\pm0.00$ | $0.00\pm0.00$ | $0.59\pm0.03$ | $0.67\pm0.12$ | $0.68\pm0.00$ | $0.26\pm0.00$ | $\mathbf{1.24\pm0.00}$ |
| | ED | $0.06\pm0.01$ | $0.08\pm0.01$ | $0.06\pm0.01$ | $0.03\pm0.01$ | $0.05\pm0.01$ | $0.05\pm0.01$ | $0.01\pm0.01$ | $0.05\pm0.02$ | $0.06\pm0.01$ | $0.04\pm0.02$ | - |
| Walker2d | UT($\times10^3$) | $1.70\pm0.01$ | $1.66\pm0.08$ | $1.49\pm0.01$ | $1.15\pm0.02$ | $0.05\pm0.00$ | $0.10\pm0.00$ | $0.41\pm0.01$ | $\mathbf{1.92\pm0.18}$ | $0.54\pm0.00$ | $0.29\pm0.01$ | $1.87\pm0.05$ |
| | HV($\times10^7$) | $4.02\pm0.01$ | $4.54\pm0.02$ | $0.03\pm0.01$ | $3.52\pm0.09$ | $0.35\pm0.00$ | $0.17\pm0.00$ | $1.22\pm0.00$ | $3.35\pm0.33$ | $1.13\pm0.00$ | $1.97\pm0.12$ | $\mathbf{4.56\pm0.00}$ |
| | ED | $0.27\pm0.03$ | $0.36\pm0.01$ | $0.00\pm0.00$ | $0.18\pm0.01$ | $0.03\pm0.01$ | $0.03\pm0.01$ | $0.15\pm0.02$ | $0.32\pm0.01$ | $0.15\pm0.02$ | $0.00\pm0.00$ | - |
| Hopper3d | UT($\times10^3$) | $1.29\pm0.02$ | $1.82\pm0.02$ | $1.25\pm0.01$ | $0.78\pm0.00$ | $0.11\pm0.00$ | $0.01\pm0.01$ | $0.03\pm0.01$ | $0.57\pm0.23$ | $1.32\pm0.01$ | $0.01\pm0.00$ | $\mathbf{2.36\pm0.01}$ |
| | HV($\times10^{11}$) | $0.92\pm0.00$ | $\mathbf{1.25\pm0.01}$ | $0.57\pm0.01$ | $0.01\pm0.00$ | $0.01\pm0.00$ | $0.00\pm0.00$ | $0.03\pm0.00$ | $0.29\pm0.00$ | $0.51\pm0.00$ | $0.00\pm0.00$ | $1.15\pm0.00$ |
| | ED | $0.05\pm0.00$ | $0.35\pm0.01$ | $0.00\pm0.00$ | $0.03\pm0.01$ | $0.00\pm0.00$ | $0.00\pm0.00$ | $0.00\pm0.00$ | $0.02\pm0.10$ | $0.10\pm0.05$ | $0.00\pm0.00$ | - |
| Ant3d | UT($\times10^3$) | $1.29\pm0.11$ | $1.81\pm0.08$ | $1.01\pm0.00$ | $0.80\pm0.00$ | $0.04\pm0.00$ | $0.05\pm0.01$ | $0.28\pm0.01$ | $1.45\pm0.10$ | $0.66\pm0.00$ | $0.01\pm0.01$ | $\mathbf{3.43\pm0.22}$ |
| | HV($\times10^{11}$) | $2.42\pm0.05$ | $2.61\pm0.02$ | $0.45\pm0.00$ | $0.21\pm0.03$ | $0.01\pm0.00$ | $0.07\pm0.01$ | $0.12\pm0.00$ | $0.73\pm0.15$ | $0.35\pm0.01$ | $0.08\pm0.00$ | $\mathbf{4.18\pm0.04}$ |
| | ED | $0.11\pm0.05$ | $0.14\pm0.04$ | $0.08\pm0.05$ | $0.08\pm0.05$ | $0.05\pm0.03$ | $0.06\pm0.03$ | $0.06\pm0.04$ | $0.17\pm0.02$ | $0.08\pm0.05$ | $0.03\pm0.02$ | - |
| Humanoid2d | UT($\times10^2$) | $-0.05\pm0.00$ | $4.51\pm0.03$ | $-0.04\pm0.00$ | $1.55\pm0.04$ | $2.71\pm0.02$ | $-3.79\pm0.72$ | $0.45\pm0.03$ | $1.43\pm0.12$ | $1.35\pm0.00$ | $2.44\pm0.00$ | $\mathbf{8.13\pm0.01}$ |
| | HV($\times10^7$) | $0.06\pm0.00$ | $1.51\pm0.04$ | $0.01\pm0.01$ | $1.01\pm0.02$ | $0.62\pm0.00$ | $0.25\pm0.01$ | $0.75\pm0.05$ | $0.77\pm0.21$ | $0.42\pm0.03$ | $1.19\pm0.05$ | $\mathbf{1.75\pm0.02}$ |
| | ED | $0.07\pm0.03$ | $0.40\pm0.04$ | $0.08\pm0.00$ | $0.07\pm0.01$ | $0.18\pm0.01$ | $0.00\pm0.00$ | $0.04\pm0.00$ | $0.15\pm0.01$ | $0.14\pm0.00$ | $0.06\pm0.00$ | - |
| Humanoid5d | UT($\times10^3$) | $0.38\pm0.01$ | $0.12\pm0.00$ | $0.08\pm0.00$ | $0.54\pm0.00$ | $0.58\pm0.00$ | $-0.10\pm0.01$ | $0.54\pm0.00$ | $0.72\pm0.02$ | $0.30\pm0.00$ | $0.55\pm0.03$ | $\mathbf{1.11\pm0.00}$ |
| | HV($\times10^{16}$) | $3.91\pm0.04$ | $1.80\pm0.02$ | $0.00\pm0.00$ | $0.59\pm0.00$ | $0.31\pm0.00$ | $0.00\pm0.00$ | $2.15\pm0.08$ | $3.48\pm0.03$ | $0.20\pm0.01$ | $4.69\pm0.20$ | $\mathbf{6.99\pm0.06}$ |
| | ED | $0.03\pm0.00$ | $0.00\pm0.00$ | $0.04\pm0.00$ | $0.10\pm0.00$ | $0.08\pm0.00$ | $0.00\pm0.00$ | $0.05\pm0.00$ | $0.09\pm0.00$ | $0.03\pm0.00$ | $0.03\pm0.00$ | - |

We also evaluated the probability of improvement (POI) between MORL-FB and the benchmark algorithms suggested by Agarwal et al. (2021) in Figure 21. POI quantifies the likelihood that one algorithm will outperform another. The results demonstrate that MORL-FB consistently achieves superior performance compared to baselines.

Additionally, we assessed the performance of training with constraint preferences using the method from Agarwal et al. (2021), presented in Figures 22 and 23.

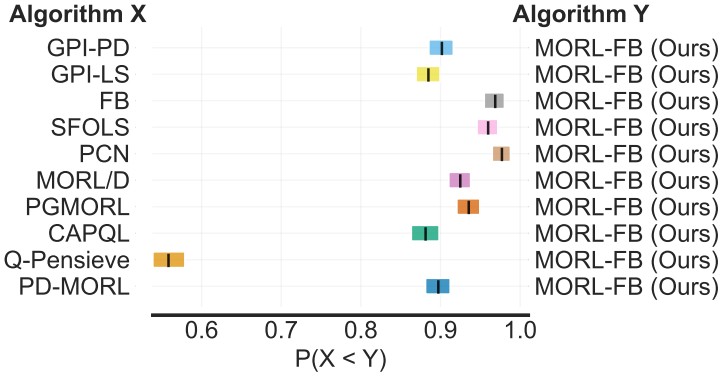

Figure 21: **Probability of Improvement (POI) of MORL-FB against benchmark algorithms.** This figure illustrates the POI of MORL-FB relative to various benchmark MORL algorithms.

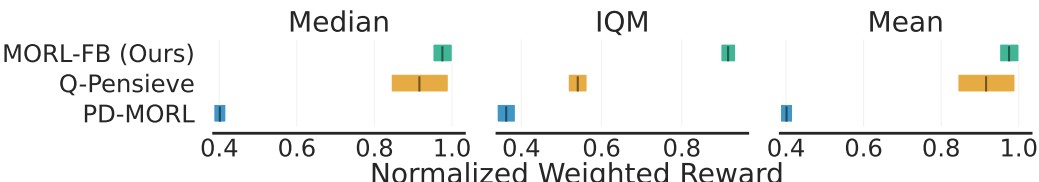

Figure 22: **Medium, IQM, and Mean performance of MORL-FB and benchmarks trained with small preference sets.** This figure displays the Median, Interquartile Mean (IQM), and Mean performance for MORL-FB and other benchmark algorithms, trained with only a small set of preference vectors.

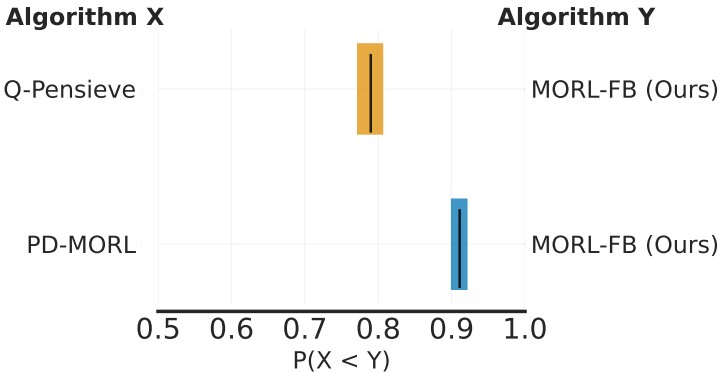

Figure 23: **Probability of Improvement (POI) of MORL-FB under constrained preference training.** This figure illustrates the POI of MORL-FB relative to other benchmark algorithms, all trained with specific constraint sets of preference vectors.

