# OpenReview forum: "A Reward-Free Viewpoint on Multi-Objective Reinforcement Learning"
_ICLR.cc/2026/Conference — ICLR 2026 Poster_

### Official Review · Reviewer_3sX6 · 2025-10-26

**Soundness:** 3
**Presentation:** 2
**Contribution:** 3
**Rating:** 6
**Confidence:** 3

**Summary:**

This paper presents a reward-free viewpoint for multi-objective reinforcement learning (MORL). It adapts the Forward–Backward (FB) representation from reward-free RL to MORL, aiming to learn a task-agnostic policy/value representation that can generalize to unseen preference vectors without retraining. Experiments on MO-Gymnasium tasks show improved performance and generalization compared with existing MORL baselines.

**Strengths:**

1. The paper provides a new perspective on MORL, suggesting that learning beyond specific scalarized reward combinations can accelerate MORL through more effective knowledge sharing across preferences.
2.The empirical evaluation is comparatively thorough for MORL: it includes multiple tasks (both discrete and continuous control), standard metrics such as hypervolume, analyses of generalization, and ablations (e.g., with and without PG-Explore and auxiliary losses).

**Weaknesses:**

1. The comparison with related work based on Successor Features (SF) and its variants is insufficient.
2. Some methodological and experimental descriptions are unclear. Please see the questions below.

**Questions:**

1. The proposed method and the goal of fast adaptation to unseen preferences appear conceptually close to SF. What concrete limitation of SF does the FB formulation address? Could the authors provide an ablation comparing FB + PG-Explore with an SF-based approach?
2. In Section 3 "Training objective function", the paper introduces both a measure loss and an auxiliary Q-loss. How are these two losses combined, particularly for the forward network? If they are combined, how sensitive is performance to their relative weighting?
3. In Figure 6, except for MORL-FB (w/o Q-loss), the hypervolume of other baselines is close to the proposed method, while UT performs significantly worse. Could the authors explain the reason for this discrepancy?

---

> ### Author Response · Authors · 2025-11-21
> **Response to Reviewer 3sX6 (1/2)**
>
> We greatly appreciate the reviewer’s insightful feedback on our work. We provide our point-by-point response as follows.
>
> **[W1]: Related work on successor features**
> > The comparison with related work based on Successor Features (SF) and its variants is insufficient.
>
> Thank you for the suggestion. In the original manuscript, our discussion of related work on RFRL and Successor Features (SF) appeared in Appendix D due to the page limit. Following the reviewer’s recommendation, and as also suggested by Reviewer cCvf, we have moved this material into the main text in the revised manuscript (see Section 5), taking advantage of the one-page extension.
>
> To further strengthen the comparison, we now dedicate a subsection (Section 5.2) specifically to SF-based methods. This subsection expands on the core ideas behind SFs, summarizes recent advances in SF variants, and discusses the key limitations that distinguish them from our RFRL-based approach. We expect that these additions provide a clearer and more comprehensive positioning of our method within the existing literature.
>
> **[Q1]: Explain the limitations of SF that FB formulation addresses**
> > The proposed method and the goal of fast adaptation to unseen preferences appear conceptually close to SF. What concrete limitation of SF does the FB formulation address?
>
> Thank you for the insightful question. While both SF-based and FB-based methods share the overarching goal of transferring knowledge across reward functions, they do so through fundamentally different mechanisms.
> SF-based methods compute feature expectations under predefined reward features, enabling fast policy evaluation for new reward weights. As a result, the primary limitations of successor features and their variants are two-fold:
>
> - (i) A pre-defined set of reward features needed: SF-based methods typically assume access to a pre-defined set of reward features. These can be manually engineered (e.g., Barreto et al., 2017) or directly derived from task rewards (e.g., Barreto et al., 2018). This dependence can limit their expressiveness and adaptability, especially in high-dimensional continuous environments where suitable reward features are non-trivial or costly to obtain.
> - (ii) Limited ability to induce optimal policies for arbitrary new reward functions: While SFs allow rapid policy evaluation for unseen reward functions, they generally do not provide a direct mechanism for computing optimal policies for arbitrary new reward functions at test time. Additional optimization or planning steps are often required, which can become computationally costly.
>
> The FB formulation addresses these limitations by adopting a reward-free MDP perspective and utilizing a low-rank decomposition of the successor measure and Q function. This approach enables the model to learn the necessary latent features directly from data, while capturing the state-action occupancy patterns of optimal policies for all reward functions within the low-rank subspace, without requiring manually specified reward features.
>
> > Could the authors provide an ablation comparing FB + PG-Explore with an SF-based approach?
>
> In the original manuscript, we included SFOLS [Alegre et al., 2022] as a representative SF-based baseline, as it applies successor features together with generalized policy improvement to MORL. Since the original SFOLS implementation targets discrete-action tasks only, we used its official code for discrete environments and extended it with a TD3 backbone for continuous-control settings. To ensure fairness, we tuned TD3-related hyperparameters (e.g., actor/critic learning rates, policy noise, and target smoothing) following the same HPO procedure used for other baselines (details in Appendix C.2).
>
> Across the full set of experiments (Figures 2, 3, and 13), MORL-FB consistently outperforms SFOLS in both UT and HV metrics, demonstrating that the FB representation combined with our proposed PG-Explore strategy can offer clear advantages over SF-based approaches for MORL.

---

> ### Author Response · Authors · 2025-11-21
> **Response to Reviewer 3sX6 (2/2)**
>
> **[Q2]: Explain how measure and Q loss are combined**
> > In Section 3 "Training objective function", the paper introduces both a measure loss and an auxiliary Q-loss. How are these two losses combined, particularly for the forward network? If they are combined, how sensitive is performance to their relative weighting?
>
> In our implementation, the measure loss and the auxiliary Q-loss are directly combined and jointly backpropagated, without tuning an additional weighting parameter. We adopted this default configuration because it already provides strong and stable performance across all evaluated tasks.
>
> To more thoroughly examine the sensitivity to the relative weighting of the two losses, we conducted an additional set of experiments in Ant where we varied the Q-loss coefficient $\alpha\_Q$ (with $\alpha\_Q = 1$ as the default). The results, summarized in the table below, indicate that MORL-FB’s overall learning performance is largely insensitive to the choice of $\alpha\_Q$, suggesting that directly combining the two losses is a robust and reasonable strategy.
>
> |Metric|Q-loss coef = 0.25|Q-loss coef = 0.5|Q-loss coef = 1.0|Q-loss coef = 2.0|
> |-|-|-|-|-|
> |UT(×$10^3$)|3.76±0.41|3.79±0.32|3.77±0.11|3.95±0.13|
> |HV(×$10^{11}$)|4.03±0.22|4.13±0.19|3.95±0.13|4.23±0.04|
>
> We have included this sensitivity analysis in Appendix D.2.5 for completeness.
>
> **[Q3]: Explain the hypervolume and utility in Figure 6**
> > In Figure 6, except for MORL-FB (w/o Q-loss), the hypervolume of other baselines is close to the proposed method, while UT performs significantly worse. Could the authors explain the reason for this discrepancy?
>
> The observed discrepancy primarily stems from the different scales used to plot UT and HV in Figure 6. In the original manuscript, HV was shown on a logarithmic scale to clearly display the behavior of “MORL-FB (w/o Q-loss),” whose hypervolume is nearly zero and therefore difficult to visualize on a linear scale. In contrast, UT was presented on a standard linear scale.
>
> To clarify this further, we provide an additional figure using a normal linear scale for HV at: https://postimg.cc/WhCtYkbp
>
> This supplementary plot confirms that the trends for both UT and HV are consistent, and that the apparent discrepancy was due entirely to the choice of visualization rather than a substantive difference in performance.

---

> > ### Comment · Reviewer_3sX6 · 2025-11-27
> >
> > Thank you for the detailed responses. My concerns have been sufficiently resolved, and I will keep my positive rating.

---

### Official Review · Reviewer_d51d · 2025-10-29

**Soundness:** 3
**Presentation:** 2
**Contribution:** 2
**Rating:** 6
**Confidence:** 4

**Summary:**

This paper proposes a reward-free viewpoint for MORL and an approach, MORL-FB, that integrates reward-free RL methods into MORL. Extensive experiments are conducted to show the effectiveness and generalization of the proposed method.

**Strengths:**

1. This work investigates the integration of a reward-free framework into multi-objective RL methods for better performance and transferability, which is novel yet natural.
2. A motivating example is provided to show the design consideration of the exploration of *z*, revealing the core insight of this work.
3. Extensive experiments are conducted on both discrete and continuous tasks, comparing with a wide range of baselines.
4. The visualizations are intuitive and show that the method is capable of handling multi-modal distributions, implying its improved generalization.

**Weaknesses:**

1. The layout on page 4 may need optimization. Algorithm 1 and Figure 1 can be presented on one line.
2. This work focuses on model design and experimental analysis, while the “reward-free viewpoint” in the title seems more theoretical. Further theoretical analysis may be needed, or the title and abstract should focus more on the model’s effect.
3. The title of Figure 20(a) seems mis-specified; it may be “Humanoid2d.”

**Questions:**

1. How is the transfer implemented through the framework of RFRL? A brief explanation based on the motivating experiment in Figure 1 would be appreciated.
2. How are $L_Q$ and $L_M$ balanced in line 20 of Algorithm 2? Is there any evidence that directly adding them up is plausible?
3. Why is the Q loss regarded as “auxiliary”? Does this mean the Q loss is less important compared with the measure loss?

---

> ### Author Response · Authors · 2025-11-21
> **Response to Reviewer d51d (1/3)**
>
> We greatly appreciate the reviewer’s insightful feedback on our work. We provide our point-by-point response as follows.
>
> **[W1,W3]: Formatting issues**
> > The layout on page 4 may need optimization. Algorithm 1 and Figure 1 can be presented on one line.
>
> We have revised page 4 to present Algorithm 1 and Figure 1 on one line, making the layout more compact and reader-friendly. Thank you for the helpful suggestion.
>
> > The title of Figure 20(a) seems mis-specified; it may be “Humanoid2d.”
>
> Thank you for catching this. We have fixed it in the updated manuscript.
>
>
> **[W2]: Refinement of title and abstract needed**
> > This work focuses on model design and experimental analysis, while the “reward-free viewpoint” in the title seems more theoretical. Further theoretical analysis may be needed, or the title and abstract should focus more on the model’s effect.
>
> Thank you for the thoughtful comment. We agree that the phrase “reward-free viewpoint” can suggest either a conceptual or a theoretical contribution. Our intention in using this term is to highlight the conceptual shift underlying our algorithm and model design: We reinterpret MORL through the lens of reward-free RL, which learns optimal policies for all possible reward functions and hence serves as a natural fit for MORL's challenge of handling unknown user preferences. This perspective directly motivates the algorithmic design of our method and distinguishes our formulation from the existing MORL methods.
>
> To address this, we have revised both the title and abstract to emphasize that our work applies a reward-free RL perspective to develop a practical MORL framework for improving the empirical data efficiency and generalization of MORL, rather than presenting formal theoretical results. Specifically:
> - Abstract: We also updated the abstract to highlight that the reward-free perspective motivates the use of RFRL’s learning objective as an auxiliary task in the design of MORL algorithms, and our contributions are primarily methodological and experimental.
> - Title: We found that during the rebuttal phase, the title cannot be updated on OpenReview. However, if title change is allowed for the camera-ready version, we would slightly revise the title as “A Reward-Free RL Framework for Multi-Objective Policy Optimization” to highlight that our focus is on the methodology and algorithm design.
>
> We appreciate the reviewer’s suggestion and believe these clarifications strengthen the presentation of the work.

---

> ### Author Response · Authors · 2025-11-21
> **Response to Reviewer d51d (2/3)**
>
> **[Q1]: Explain the transfer implemented through RFRL**
> > How is the transfer implemented through the framework of RFRL? A brief explanation based on the motivating experiment in Figure 1 would be appreciated.
>
> We respectfully interpret the reviewer’s question “*How is the transfer implemented through the framework of RFRL?*” as asking how the RFRL framework described in Section 3.1 enables transfer across reward functions (i.e., across preferences in MORL), and how this is illustrated in the motivating Deep Sea Treasure (DST) example in Figure 1.
>
> If this interpretation is not what the reviewer intended, we sincerely apologize and would be grateful for clarification.
>
> Below we provide a concise explanation of the training and testing processes in the FB-based RFRL framework and how transfer arises therein.
>
> **1. Training (Reward-Free Phase)**
>
> The objective of FB-based RFRL is to learn representations that allow computing (near-)optimal Q-values for any scalar reward function. In the DST example, this includes all linear scalarizations corresponding to all preferences.
>
> **(i) Encoding reward functions:**
> Each scalar reward function $R$ is mapped to a latent vector $z_R$, and a $z_R$-conditioned Q-network $Q(s,a,z_R)$ is trained. In DST, each preference yields a different scalarized reward and hence a different $z_R$.
>
> **(ii) Low-rank decomposition of the Q-function:**
> FB formulation decomposes the Q-function as:
>
> $$
>  Q(s,a,z_R)=\mathbf{F}\_\theta(s,a,z_R)^\top z_R,
>  \qquad
>  z_R = \mathbb{E}\_{(s,a)\sim\mathcal{D}}[\mathbf{B}\_\omega(s,a) R(s,a)],
> $$
>
> where $\mathbf{F}\_\theta$ and $\mathbf{B}\_\omega$ together approximate the successor measure of the optimal policy for any $R$. In DST, this corresponds to learning the visitation patterns of optimal policies for all preferences.
>
>
> **(iii) Training losses:**
> The measure loss enforces Bellman consistency of successor measures.
> In MORL-FB, we additionally include an auxiliary Q-loss using observed vector rewards to help train $\mathbf{F}_\theta$.
>
> In DST, as the two-dimensional rewards (i.e., treasure value and step cost) can be observed, we can easily calculate the auxiliary Q-loss under different preferences. Meanwhile, the measure loss can be directly computed by the collected (reward-free) state-action pairs.
>
> **(iv) Preference-guided exploration (PG-Explore):**
> To support transfer across reward functions, the model must learn representations for many possible $z_R$. In MORL-FB, we introduce the PG-Explore strategy to sample latent vectors relevant to MORL reward structures.
>
> In DST, this involves forming $\hat{z}\_\lambda$ using sampled mini-batches and preference-weighted rewards:
>
> $$
> \hat{z}\_\lambda = \frac{1}{|\mathcal{D}|}\sum\_{(s,a,\mathbf{r},s')\in\mathcal{D}} \mathbf{B}_\omega(s,a)\mathbf{r}^{\top}\lambda.
> $$
>
> This ensures effective transfer across preferences during learning.
>
>
>
> **2. Testing (Post-Reward Phase)**
>
> Given a test preference $\lambda_{\text{test}}$, we convert the vector reward into a scalarized reward as $R(s,a) \leftarrow \lambda_{\text{test}}^\top \mathbf{R}(s,a)$, compute the corresponding latent representation $z_R$, and obtain the induced policy as follows (cf. Equation (3) in the paper):
>
> $$
> \pi(s,z_R) = \arg\max_a \mathbf{F}_\theta(s,a,z_R)^\top z_R.
> $$
>
> This retrieves an optimal policy for the specified test-time preference, demonstrating transfer from the reward-free phase to downstream MORL tasks.
>
> We hope this explanation addresses the reviewer’s question regarding the notion of transfer within the RFRL framework and its illustration in the DST example. If our interpretation of the question is not completely accurate, we would be grateful for further clarification so we can address the question more precisely.

---

> ### Author Response · Authors · 2025-11-21
> **Response to Reviewer d51d (3/3)**
>
> **[Q2]: Explain the balance of measure loss and Q loss**
> > How are $L_Q$ and $L_M$ balanced in line 20 of Algorithm 2? Is there any evidence that directly adding them up is plausible?
>
> In our implementation, the measure loss $\mathcal{L}_M$ and the Q loss $\mathcal{L}_Q$ are directly added together without an additional weighting term. We chose this default configuration because it already yields strong and stable performance across all evaluated tasks, and we did not observe a need for further hyperparameter tuning during development.
>
> Moreover, as suggested by Reviewer 3sX6, we have conducted an additional sensitivity analysis to better understand how the balance between these two losses affects performance. Specifically, we varied the Q-loss coefficient $\alpha_Q$ (with $\alpha_Q=1$ as the default setting) and reported the corresponding results in Ant3d. The experimental results show that MORL-FB’s performance is largely insensitive to the choice of $\alpha_Q$, indicating that directly adding $\mathcal{L}_M$ and $\mathcal{L}_Q$ is a reasonable and robust design choice.
>
> |Metric|Q-loss coef = 0.25|Q-loss coef = 0.5|Q-loss coef = 1.0|Q-loss coef = 2.0|
> |-|-|-|-|-|
> |UT(×$10^3$)|3.76±0.41|3.79±0.32|3.77±0.11|3.95±0.13|
> |HV(×$10^{11}$)|4.03±0.22|4.13±0.19|3.95±0.13|4.23±0.04|
>
>
> We have included these sensitivity results in Appendix D.2.5 for completeness.
>
> **[Q3]: Explain why Q loss is regarded as auxiliary**
> > Why is the Q loss regarded as “auxiliary”? Does this mean the Q loss is less important compared with the measure loss?
>
> Both the measure loss and the Q loss play essential roles in the effectiveness of MORL-FB: (1) The measure loss provides the core training objective of an FB-based RFRL method and forms the foundation of the learned representation. (2) At the same time, the Q loss contributes substantially to performance by facilitating more effective policy learning, as evidenced by our ablation study (Figure 6).
>
> Our use of the term “auxiliary” is not intended to imply that the Q loss is less important, but rather to clarify its relationship to the FB framework. FB-based RFRL methods are fundamentally constructed around the reward-free measure loss. Therefore, from a representation-learning standpoint, the Q loss is auxiliary in the sense that it complements and enhances the successor-measure-based objective, rather than serving as the primary FB learning signal. In practice, however, both losses are integral to the full MORL-FB algorithm.

---

> > ### Comment · Reviewer_d51d · 2025-11-27
> >
> > I thank the authors for their detailed response and revisions. My concerns have been fully addressed, and the formatting improvements have enhanced the paper's readability.
> >
> > I believe this work effectively integrates RFRL with Multi-Objective RL, achieving a generalized MORL approach through the reward-free framework. I maintain my positive assessment.

---

### Official Review · Reviewer_hbj3 · 2025-10-30

**Soundness:** 4
**Presentation:** 3
**Contribution:** 3
**Rating:** 8
**Confidence:** 4

**Summary:**

This paper views MORL problems from a reward-free perspective and then solves them using the RFRL method. Based on RFRL, it introduces a preference-guided exploration strategy that incorporates reward functions and preferences into the original RFRL. Specifically, rather than directly sampling latent vectors from a normal distribution, it samples preferences uniformly and uses them to generate latent vectors. Finally, the proposed method is evaluated on a series of tasks in MO-Gymnasium and demonstrates superior performance and data efficiency.

**Strengths:**

- The paper is well-motivated. It solves MORL problems via RFRL, which is intuitive and novel and could serve as a good bridge between the two sub-fields.
- The empirical evaluation and ablation studies are comprehensive and insightful, carefully demonstrating the effectiveness of the proposed approach from multiple aspects.
- The paper is well-written and well-organized, with implementation details that are easy to understand.

**Weaknesses:**

I didn’t find major issues in this paper.

**Questions:**

- For completeness, I recommend briefly describing the future visited probabilities $M=FB$ in the preliminary section or appendix, as it is an important concept that helps readers understand why FB can be applied to arbitrary rewards.
- In Figure 1, the lines are too dense and intertwined, making it difficult to interpret. It may be better to eliminate the curves corresponding to batch sizes 256 and 4096, or adjusting the scaling to improve clarity.
- I recommend adopting more RFRL methods beyond FB to solve MORL in the future, which could lead to more insightful conclusions on RFRL's advantages in MORL.

---

> ### Author Response · Authors · 2025-11-21
> **Response to Reviewer hbj3**
>
> We thank the reviewer for the encouraging and insightful feedback on our work. We provide our point-by-point response as follows.
>
> **[Q1]: Provide more background on successor measure**
> > For completeness, I recommend briefly describing the future visited probabilities in the preliminary section or appendix, as it is an important concept that helps readers understand why FB can be applied to arbitrary rewards.
>
> We thank the reviewer for the helpful suggestion regarding completeness. We agree that a brief description about the successor measure is helpful for readers to understand how the FB method can be applied to arbitrary reward functions. To ensure the paper is fully self-contained and clear, we have added a new appendix section, "Additional Background: Successor Measure and Forward-Backward Representations," in Appendix A to provide the necessary context about successor measure and its connection with the FB representations.
>
> **[Q2]: Improve the clarity of Figure 1**
> > In Figure 1, the lines are too dense and intertwined, making it difficult to interpret. It may be better to eliminate the curves corresponding to batch sizes 256 and 4096, or adjusting the scaling to improve clarity.
>
> We have adjusted the position of the legend and increased the font size to make Figure 1 easier to read. Thank you for your feedback.
>
> **[Q3]: Exploring more RFRL methods beyond FB for MORL as a future direction**
> > I recommend adopting more RFRL methods beyond FB to solve MORL in the future, which could lead to more insightful conclusions on RFRL's advantages in MORL.
>
> We totally agree that leveraging more RFRL methods beyond FB representation, such as representation learning methods with successor measures [Agarwal et al., 2025; Farebrother et al., 2023] and learning distance-preserving state representations [Park et al., 2024], to address the MORL problem is a promising direction for offering more insight into how MORL could benefit from RFRL. We have also included these future research directions in Section 6 in the updated manuscript.
>
> [Agarwal et al., 2025] Siddhant Agarwal, Harshit Sikchi, Peter Stone, and Amy Zhang, “Proto successor measure: Representing the behavior space of an RL agent,” ICML 2025.
>
> [Farebrother et al., 2023] Jesse Farebrother, Joshua Greaves, Rishabh Agarwal, Charline Le Lan, Ross Goroshin, Pablo Samuel Castro, and Marc G. Bellemare, “Proto-value networks: Scaling representation learning with auxiliary tasks,” ICLR 2023.
>
> [Park et al., 2024] Seohong Park, Tobias Kreiman, and Sergey Levine, “Foundation policies with Hilbert representations,” ICML 2024.

---

> > ### Comment · Reviewer_hbj3 · 2025-11-27
> >
> > Thank you for the responses. They address my comments, and I maintain my original rating.

---

### Official Review · Reviewer_cCvf · 2025-11-01

**Soundness:** 3
**Presentation:** 3
**Contribution:** 3
**Rating:** 8
**Confidence:** 3

**Summary:**

This paper considers the problem of multi-objective reinforcement learning. Current MORL algorithms typically assign preference weights to individual objectives during the training process, modifying the underlying utility function to discover one or multiple policies to satisfy different preferences.
This work proposes to instead use techniques adapted from reward-free reinforcement learning (RFRL), where no such preference weighting is included in the utility function. RFRL aims to learn a policy that is optimal for arbitrary reward functions. The authors note the conceptual connection to MORL problems, and suggest an extension of a state-of-the-art RFRL algorithm to the multi-objective setting. The proposed algorithm contains three important modifications: preference-guided exploration, training on latent vectors sampled from the replay buffer as auxiliary tasks, and auxiliary Q loss.
The algorithm is validated experimentally, showing clear improvement over other MORL algorithms in several standard multi-objective metrics.

**Strengths:**

- This paper draws a natural connection between RFRL and MORL, with a generally insightful discussion of the similarities and differences between the two. According to the authors, this is the first work to do so explicitly. This is a valuable contribution, and an interesting starting point for further work. (This reviewer is not intimately familiar with the current state of the art, and so must take the assertions about the novelty of the paper at face value)

- The paper is well-written and largely well-organised. Experiments are well-constructed, with a solid statistical approach, hyperparameter search, reproducible parameters, and meaningful ablation studies.

- The experimental validation of the algorithm shows good results compared to various state-of-the-art approaches and baselines.

**Weaknesses:**

- The paper occasionally seems to conflate the field of MORL as a whole with the single MORL technique of linearly combining objectives, e.g. in the introduction and Section 3. As mentioned in the related work section, other MORL approaches exist. This should be clarified.

- At times, this paper relies quite heavily on the appendix, in particular when relegating the discussion of related work from RFRL to the appendix. With the extension of the page limit during the revision phase, this should be moved to the main section to allow the paper to stand alone.

- The conclusion in its current form is also quite short, and could benefit from the addition of slightly more detail about the method and context. Similarly, some figures are quite cramped (see additional comments).


Additional (minor) comments

- Some of the figures, e.g. 1 and 3, are somewhat difficult to read. These could be made more readable despite the space constraints, e.g. by increasing font size and line thickness. Similarly, it is good that variance/error is represented in the plots, but in the current format this is barely readable.

- Some minor language and formatting issues:
1. line 050: “no prior work has explicitly adapt RFRL...”
2. “across various tasks in … benchmark” (line 080/081)
3. line 210: “where (a) holds by that ...”
4. Eq. 5 extends into the page margin.

- Occasionally, a citation is used as the subject of a sentence, e.g. in line 465 “(Felten et al., 2024; Mossalam et al., 2016) extended …”. In such cases, ‘\citet{}’ can be used in LaTeX to generate a textual citation.

**Questions:**

1. Perhaps this has been missed, but will the implementation of this methods be made available, ideally for integration with the existing benchmarking suite?

2. How exactly were hyperparameters chosen for the proposed method (MORL-FB)? The appendix only mentions a decision “guided by prior research and the HPO results”, which seems not quite comparable to the HPO-tuning of the remaining algorithms.

---

> ### Author Response · Authors · 2025-11-21
> **Response to Reviewer cCvf (1/2)**
>
> We greatly appreciate the reviewer’s thoughtful and encouraging feedback on our work. We provide our point-by-point response as follows.
>
> **[W1, W3]: Regarding the presentation**
> > The paper occasionally seems to conflate the field of MORL as a whole with the single MORL technique of linearly combining objectives, e.g. in the introduction and Section 3. As mentioned in the related work section, other MORL approaches exist. This should be clarified.
>
> We appreciate the reviewer’s careful reading and valuable suggestions. We recognize the potential conflation in our initial presentation, and we have carefully revised the language in the Introduction and Section 3 to better contextualize our methodology. Thank you again for the helpful feedback, which has helped us improve the clarity of our discussion.
>
> > The conclusion in its current form is also quite short, and could benefit from the addition of slightly more detail about the method and context.
>
> We have addressed this by expanding the conclusion section (Section 6) in two ways: (i) We provide more context about the RFRL perspective of MORL and the proposed design of MORL-FB to highlight the main contributions. (ii) Additionally, following the suggestion from Reviewer hbj3, we further discuss the promising future directions of exploring broader RFRL methods beyond the FB formulation, such as other representation learning methods of successor measure and learning distance-preserving state representations.
>
>
> **[W2]: Related work about RFRL**
> > At times, this paper relies quite heavily on the appendix, in particular when relegating the discussion of related work from RFRL to the appendix. With the extension of the page limit during the revision phase, this should be moved to the main section to allow the paper to stand alone.
>
> Thank you for the helpful suggestion. We agree that relocating the discussion of RFRL-related work from the appendix to the main text will make the paper more self-contained. In addition, as Reviewer 3sX6 recommended expanding the discussion of related work on successor features (SF), we took this opportunity to reorganize and strengthen the entire related-work section.
>
> Accordingly, we have moved the material previously in Appendix D into Section 5 and improved the structure by introducing dedicated subsections for SF (Section 5.2) and RFRL (Section 5.3). We expect that this restructuring makes the relevant context immediately accessible and easier for readers to follow.
>
>
>
>
> **[W4]: Additional comments on the formatting**
>
> > Some of the figures, e.g. 1 and 3, are somewhat difficult to read.
>
> We have adjusted the position of the legend and increased the font size of Figures 1 and 3 accordingly to make them cleaner and easier to read. Thank you for your feedback.
>
> > There are some minor language and formatting issues.
>
> Thank you for catching these. We have fixed them in the updated manuscript.
>
> > Occasionally, a citation is used as the subject of a sentence, e.g. in line 465 “(Felten et al., 2024; Mossalam et al., 2016) extended …”. In such cases, ‘\citet{}’ can be used in LaTeX to generate a textual citation.
>
> We have done a thorough check through the paper and fixed them in the updated manuscript. Thank you for the suggestion.

---

> ### Author Response · Authors · 2025-11-21
> **Response to Reviewer cCvf (2/2)**
>
> **[Q1]: Implementation to be made publicly available**
> > Perhaps this has been missed, but will the implementation of this method be made available, ideally for integration with the existing benchmarking suite?
>
> Thank you for the suggestion. Yes, the full implementation of our method is included in the supplementary material provided with this submission. We also plan to release our GitHub repository upon acceptance of the paper to ensure easy access and reproducibility. For detailed descriptions of parameters and configurations, please refer to Appendix C.2 in the updated manuscript (Appendix B.2 in the original submission).
>
> Regarding integration with an existing benchmarking suite such as MORL-Baselines ([https://github.com/LucasAlegre/morl-baselines](https://github.com/LucasAlegre/morl-baselines)), we agree that this would be highly valuable for the community. This integration is feasible because MORL-Baselines is designed to accommodate different MORL agents within a unified framework. In particular:
> - We would be able to reuse several shared components, such as the policy network architecture and the standard replay buffer, provided in the multi-policy module ([https://github.com/LucasAlegre/morl-baselines/tree/main/morl_baselines/multi_policy](https://github.com/LucasAlegre/morl-baselines/tree/main/morl_baselines/multi_policy)).
> - MORL-FB also introduces several components not currently present in MORL-Baselines, such as the forward and backward networks used for FB representation learning. These can be integrated by adding a dedicated implementation file in a separate folder under the same multi-policy directory, following the structure used for other MORL algorithms in the suite. We plan to make this integration available after acceptance.
>
> **[Q2]: Hyperparameters of MORL-FB**
> > How exactly were hyperparameters chosen for the proposed method (MORL-FB)? The appendix only mentions a decision “guided by prior research and the HPO results”, which seems not quite comparable to the HPO-tuning of the remaining algorithms.
>
> The hyperparameters of MORL-FB were selected through a combination of hyperparameter optimization (HPO) and default choices inherited from the underlying base algorithms. Specifically:
> - The hyperparameters tuned via HPO are listed in Table 5 (Appendix C.2). These were chosen using exactly the same HPO procedure as applied to all baseline methods, ensuring a fair and comparable tuning process.
> - The additional hyperparameters reported in Table 12 were adopted directly from the recommended default settings of the original FB and TD3 algorithms. We chose not to further tune these parameters because the default configurations already yielded strong empirical performance in our preliminary experiments, and additional tuning did not appear necessary.
>
> We have updated Appendix C.2 to clearly explain this selection process.

---

### Author Response · Authors · 2025-11-21
**Response to All Reviewers**

We sincerely thank all the reviewers for their insightful comments and constructive suggestions, which have significantly helped us improve the clarity and presentation of our paper. In response to the feedback, with the one-page extension, we have revised the paper and made several improvements to enhance readability, clarity, and overall presentation, with changes highlighted in blue. We summarize the key updates as follows.
### **(1) Layout and figure adjustments for visual clarity**
Following Reviewer d51d’s suggestion, we improved the layout of Algorithm 1 and Figure 1 to ensure a cleaner visual structure. Additionally, in response to Reviewers cCvf and hbj3, we refined the presentation of Figures 1 and 3 to further enhance their readability and visual clarity.

### **(2) Refined Related Work**
Based on the comments from Reviewers cCvf and 3sX6, given the one-page extension, we have moved the related work about RFRL and successor features (SF) to the main text (Sections 5.2 and 5.3) and expanded it to more clearly describe their connections and strengthen the contextual framing of our contributions.

### **(3) Refined abstract**
In response to Reviewer d51d, we have refined the abstract to better highlight the algorithmic and experimental focus of MORL-FB as well as the model’s empirical strengths, ensuring a more coherent presentation of the paper’s main contributions.

### **(4) Additional background on successor measure**
Following Reviewer hbj3’s suggestion, we added one appendix section (cf Appendix A) to provide the context about the basic concept of successor measure and its connection with the FB representations.

### **(5) Conclusion augmentation**
As recommended by Reviewers cCvf and hbj3, we expanded Section 6 (Conclusion, Limitations, and Future Work) to provide a more complete and coherent closing discussion.

---

### Comment · Area_Chair_d9Va · 2025-11-25

Dear Reviewers

Thank you for your time and help for reviews.
The author-reviewer discussion due is in one week. If you have not done yet, please review the authors' rebuttal for the paper under your evaluation and engage in discussion with authors.

Thank you again.
Best,

Area Chair

---

### Meta-Review · Area_Chair_2Cx2 · 2026-01-07

**Summary:**

This submission proposes a novel connection between reward-free reinforcement learning (RFRL) and multi-objective reinforcement learning (MORL), instantiating it via an adapted FB-style RFRL method with preference-guided exploration and auxiliary training signals. Reviewers found the idea well-motivated and the empirical study strong (broad MO-Gymnasium evaluation, meaningful ablations, solid reproducibility), with clear improvements over competitive MORL baselines. The main issues raised during review were primarily about clarity/positioning (e.g., properly situating the method within broader MORL beyond linear scalarization, strengthening the related-work discussion, and improving presentation/figures) rather than technical soundness.

**Reviewer Concerns:**

Based on the rebuttal and subsequent discussion, all reviewer concerns have been satisfactorily addressed:

- **Positioning within broader MORL** (beyond linear scalarization): clarified with improved contextualization.
- **Related-work completeness** (e.g., Successor Features, RFRL background): expanded and integrated into the main paper.
- **Missing background for key concepts:** authors added additional explanations and a dedicated background section.
- **Methodological clarifications** (loss weighting, hyperparameter fairness, metric interpretation): adequately explained, with added sensitivity analysis where needed.
- **Presentation issues** (figure readability, layout, minor typos): fixed in the revised version.
- **Reproducibility:** authors clarified availability of code and intend to release a public repository.

All reviewers indicated that their concerns were resolved and did not raise any remaining issues.

**Reviewer Scores:**

Given the rebuttal and the reviewers’ post-rebuttal messages, I expect scores would remain unchanged after full discussion. The discussion consolidates the existing positive evaluations rather than materially shifting them.

---

### Decision · Program_Chairs · 2026-01-26

Accept (Poster)